# CheXGenBench: A Unified Benchmark For Fidelity, Privacy and Utility of Synthetic Chest Radiographs

**Raman Dutt**[1]                                    *raman.dutt@ed.ac.uk*
**Pedro Sanchez**[1]                               *pedropaulosg@hotmail.com*
**Yongcheng Yao**[1]                                   *yc.yao@ed.ac.uk*
**Steven McDonagh**[1]                             *s.mcdonagh@ed.ac.uk*
**Sotirios A. Tsaftaris**[1]                         *S.Tsaftaris@ed.ac.uk*
**Timothy Hospedales**[1,2]                        *t.hospedales@ed.ac.uk*
[1] *University of Edinburgh*
[2] *Samsung AI Center, Cambridge*

**Reviewed on OpenReview:** *https://openreview.net/forum?id=wrKmzYQACp*

## Abstract

Structured benchmarks have advanced text-conditional image generation for real-world imagery, however, no such benchmark exists for synthetic radiograph generation. Despite being a highly active area of research, existing studies continue adopting inconsistent evaluation protocols and lack a unified assessment of the three most critical criteria: generative fidelity, privacy risk, and downstream utility. To address these limitations, we introduce CheXGenBench, the ***first*** unified evaluation framework for synthetic chest radiograph generation that simultaneously assesses fidelity, privacy risks, and downstream utility across frontier text-to-image (T2I) generative models. Our evaluation protocol, comprising over 20 quantitative metrics, covers 11 leading T2I architectures with *plug-and-play* integration for newer models. Through a rigorous and fair evaluation protocol, we establish comprehensive baseline state-of-the-art (SoTA) performances across all dimensions to guide future research. Furthermore, our results uncover several limitations of current generative models, which include **first**, even SoTA models struggle with long-tailed medical distributions; **second**, models pose high privacy risks regardless of fidelity quality; and **third**, while synthetic data already benefits downstream classification, it is of limited utility for downstream multimodal tasks. Drawing from these results, we propose concrete research directions to advance the field. The code is available at this URL.

## 1 Introduction

Recent advances in multi-modal generative models, particularly Text-to-Image (T2I) systems (Ramesh et al., 2021; Saharia et al., 2022; Hurst et al., 2024; Dutt et al., 2025b; Xie et al., 2025), have demonstrated remarkable capabilities in producing high-fidelity synthetic images that closely adhere to natural-language prompts. Central to this progress is the development of comprehensive, well-designed benchmarks that evaluate various aspects of generative performance. These benchmarks drive innovation by establishing standardised evaluation protocols and creating an equitable foundation for model comparison. The natural imaging domain has benefited from numerous such benchmarks, each meticulously assessing specific aspects and identifying limitations that researchers subsequently address in developing next-generation models. For example, MS-COCO dataset (Lin et al., 2014) has been established as a seminal benchmark for evaluating general performance across multiple tasks, with particular emphasis on text-guided image generation. Building upon this foundation, more specialised benchmarks have emerged to assess specific attributes such as compositional understanding (Huang et al., 2023; 2025; Ghosh et al., 2023), enabling more nuanced analysis of model capabilities. Despite the advancements in the natural imaging domain, there remains a significant gap in

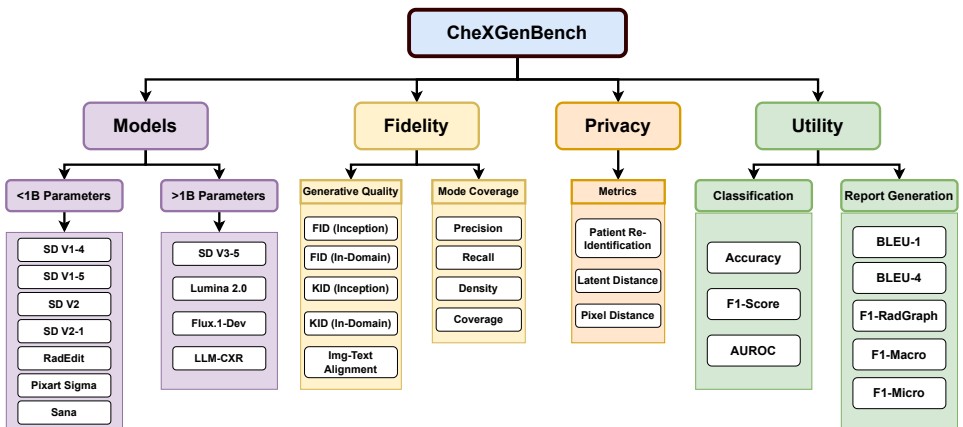

Figure 1: Figure illustrating the overall schematic of the CheXGenBench benchmark for evaluating text-to-image models in synthetic radiograph generation. CheXGenBench organises evaluation into three dimensions: **Fidelity** (measuring generative quality and mode coverage), **Privacy** (assessing memorisation and re-identification risks), and **Utility** (evaluating practical value of synthetic samples through image classification and radiology report generation metrics) through 20+ metrics across 11 widely-adopted **T2I models.**

medical image analysis, and specifically in terms of benchmarking specialised tasks such as text-to-image generation.

**Benchmarking Medical Imaging AI:** Medical image analysis has made significant strides by benefiting from rapid advancements in artificial intelligence; however, its progress faces substantial constraints due to what has been characterised as a benchmarking crisis (Mahmood, 2025). The ultimate goal of AI applications in medicine remains the development of intelligent systems capable of supporting and potentially transforming clinical decision-making processes. Progress toward this objective is frequently hindered by insufficient transparency in training and evaluation protocols, coupled with fragmented assessment criteria. Consequently, claims of "*state-of-the-art*" performance are often premature, non-reproducible, or reflect narrow contextual improvements rather than demonstrating genuine translational capability to clinical practice. The challenges of benchmarking in medical AI diagnostic tasks such as classification, localisation, segmentation and report generation have been discussed, and some improvements have been proposed (Irvin et al., 2019; Zong et al., 2023; Karargyris et al., 2023; Dutt et al., 2024a; Zhang et al., 2024). However, despite opportunities for impactful societal applications from diagnostic training to rare condition simulation, benchmarking in medical image *generation* remains in an even more nascent and unsatisfactory state. This is exacerbated by unique challenges such as unclear evaluation metrics, data scarcity due to privacy constraints, and long-tailed distributions of rare pathologies.

While medical AI diagnostic tasks have been long studied, the generation of synthetic data through text prompts (medical descriptions) has recently emerged as a critical research focus. Besides being an interesting academic measure of medical AI capability, the ultimate motivation is that progress in diagnostic tasks is usually bottlenecked by *data scarcity* in the medical domain. High-fidelity generative models for synthetic data offer the promise to alleviate this bottleneck and ultimately facilitate clinical impact (Giuffrè & Shung, 2023). Chest radiographs are the most commonly used frontline modality in medical imaging. Although the majority of collected clinical data remain inaccessible behind institutional firewalls and compliances (Mahmood, 2025), several data repositories have been opened (Johnson et al., 2016; 2019; Irvin et al., 2019; Dutt & Krishna, 2019; Zhang et al., 2025), facilitating the development of generative models capable of synthesizing radiographs with potential relevance to downstream clinical utility (Ghalebikesabi et al., 2023; Bluethgen et al., 2024; Lee et al., 2024).

Current research on Text-to-Image generation of radiographs can be broadly categorized into two primary streams: **(1)** studies prioritizing the enhancement of image fidelity and generative performance (Lee et al.,

2023; Weber et al., 2023; Lee et al., 2024; Dutt et al., 2024b; Bluethgen et al., 2024; Han et al., 2024; Huang et al., 2024; Morís et al., 2024), and **(2)** investigations focusing on the mitigation of privacy concerns and patient re-identification risks (Fernandez et al., 2023; Dar et al., 2023; Akbar et al., 2025; Dutt et al., 2025a; Dutt, 2025) that could undermine synthetic data utility. Despite being an active area of research with significant contributions, we have identified several critical benchmarking dimensions in which existing studies demonstrate consistent limitations.

1. **Minimal or Absent Comparative Baselines:** Several notable studies either include minimal (self-proposed) (Dutt et al., 2024b; 2025a) or no baselines (Weber et al., 2023) for evaluation. Han et al. (2024); Lee et al. (2024) limit their comparative evaluations to only two competing methodologies.

2. **Reporting Inadequate Metrics:** The existing literature predominantly reports Fréchet Inception Distance (FID) (Heusel et al., 2017), often computing it with in-domain, medical image encoders based on primitive architectures (Cohen et al., 2022), failing to reliably estimate fidelity. Furthermore, no studies adequately report metrics characterising mode-coverage, a criterion of paramount importance in synthetic image generation for capturing the diversity of the underlying data distribution.

3. **Reporting Global Averages:** All metrics in existing literature are reported as micro-averages across entire datasets disregarding the long-tailed nature of medical data distributions. For example, excellent *average* generation fidelity could be reflective of the ability to generate more common images of healthy individuals, and the inability to generate images with pathologies of interest. This situation would offer limited clinical utility for pathology diagnosis.

4. **Restriction to Early-Stage Architectures:** Existing studies (Bluethgen et al., 2024; Weber et al., 2023; Dutt et al., 2024b; Favero et al., 2025) have predominantly confined themselves to outdated T2I model architectures (Rombach et al., 2022), failing to address the crucial question of how recent advancements in generative modelling (Chen et al., 2024; Xie et al., 2025; Labs, 2024) from the natural imaging domain translate to the specialised requirements of medical imaging contexts.

5. **Lack of a Unified Evaluation:** Existing studies are fragmented between evaluating generative fidelity (Bluethgen et al., 2024; Weber et al., 2023; Lee et al., 2024) or privacy and re-identification risks (Fernandez et al., 2023; Akbar et al., 2025; Dutt, 2025), failing to conduct and provide a unified evaluation of the two key aspects of synthetic radiograph generation research.

6. **Limited Evaluation on Synthetic Data Utility:** Most studies fail to comment on the downstream utility of synthetic radiographs (Weber et al., 2023; Lee et al., 2024) often presenting generation results (FID scores) without rigorously assessing their potential impact on medical image analysis tasks such as classification, segmentation, or diagnostic reasoning. This need for rigour is reflected in recent standardisation efforts, such as the "Scorecards" for synthetic medical data proposed by (Zamzmi et al., 2024), which reports on key dimensions like fidelity, utility, and privacy.

To address these limitations, we introduce CheXGenBench, a comprehensive benchmark designed for rigorous evaluation of frontier generative models across a diverse spectrum of metrics encompassing: **(1)** generation fidelity and mode coverage, **(2)** privacy and re-identification risk assessment, and **(3)** downstream clinical utility through an extensive suite of 20+ quantitative and interpretable metrics. We establish standardised training and evaluation protocols to enable equitable comparison between diverse model architectures. Furthermore, our work is highly complementary to standardised evaluation approaches such as model and data card (Zamzmi et al., 2024) by providing the information required to populate those cards. CheXGenBench prioritises usability and adaptability, facilitating seamless plug-and-play integration of both existing and emerging generative frameworks. Through systematic evaluation, we present several T2I models previously **unexplored** for radiograph generation and establish new state-of-the-art (SoTA) performance.

## 2 CheXGenBench Design

CheXGenBench is designed to evaluate each text-to-image model across three dimensions comprehensively:
**(1)** generative fidelity and mode coverage (Sec 2.2), **(2)** privacy and patient re-identification risks (Sec 2.2),
and **(3)** synthetic data utility for downstream tasks (Sec 2.4). We elucidate these in the following sub-sections.

**Design Principles:** To maximise usability, CheXGenBench features *decoupled* training and evaluation
pipelines. This allows researchers to use their preferred training frameworks (e.g., Hugging Face Diffusers
(von Platen et al., 2022)) and then automatically assess their models on over 20 standardized metrics by
simply providing the generated images and a metadata file.

### 2.1 Training Dataset and Protocol

**Criteria for Model Inclusion:** Our model selection was guided by two main criteria. First, we included
models previously used for synthetic radiograph generation in existing literature (Rombach et al., 2022;
Bluethgen et al., 2024; Lee et al., 2024; Pérez-García et al., 2024) to provide continuity and comparability
with prior work. Second, we incorporated recent state-of-the-art models (both diffusion and auto-regressive)
(Chen et al., 2024; Esser et al., 2024; Xie et al., 2025; Qin et al., 2025; Labs, 2024) from the natural imaging
domain that had not yet been evaluated for chest X-ray synthesis. Thus we both benchmark established
methods and assess the potential of newer architectures for medical image generation.

The models in our benchmark were stratified into two categories based on parameter count: (1) models with
fewer than 1B parameters, and (2) models exceeding 1B parameters. For the smaller models, we employed
full fine-tuning (FFT) of all parameters. For larger architectures, we implemented Low-Rank Adaptation
(LoRA) (Hu et al., 2022) with rank 32 on standard query, key, and value layers in the attention blocks to
address both computational constraints and reflect realistic training scenarios. In Appendix A.6, we present
an ablation study that investigates the impact of increasing the LoRA rank for these models.

**Training Dataset with Enhanced Captions:** All evaluations are conducted on the MIMIC-CXR dataset
(Johnson et al., 2019), which has become the de-facto standard for text-to-image radiograph generation
(Bluethgen et al., 2024; Weber et al., 2023; Pérez-García et al., 2024; Lee et al., 2024; Dutt et al., 2024b).
While prior work has often relied on abbreviated captions derived from rule-based methods (Pérez-García
et al., 2024; Bluethgen et al., 2024), these approaches can be inadequate due to inconsistencies in clinical
terminology and report structure (Zambrano Chaves et al., 2025). Reflecting a recent shift towards deep
learning techniques for generating more comprehensive summaries (Segalis et al., 2023; Zambrano Chaves
et al., 2025), and motivated by evidence that more descriptive captions enhance generative fidelity, we are
the first study to adopt the enhanced "LLaVA-Rad" annotations for this task. We empirically validate this
choice in Appendix A.9, where we demonstrate that using these annotations leads to improved image fidelity
and reduced re-identification risks, and thus we strongly recommend them for future research in this domain.

**Training Protocol:** Our analysis revealed that prior studies employed inconsistent training budgets,
undermining valid cross-model comparison. To establish a level evaluation framework, we implemented a
standardised training protocol across all T2I models. Each architecture was trained for precisely 20 epochs
on an identical training split of 237,388 samples annotated with "LLaVA-Rad" annotations. We release the
training and evaluation data splits along with the benchmark.

### 2.2 Evaluating Generative Fidelity, Mode Coverage, and Conditional Analysis

**Limitations of Current Fidelity Assessment:** The Fréchet Inception Distance (FID) score (Heusel et al.,
2017) is widely used for evaluating synthetic radiograph fidelity. Standard FID uses InceptionV3 (Szegedy
et al., 2016) features trained on natural images, creating domain mismatch for medical images (Kynkäänniemi
et al., 2023). While some research employs domain-specific encoders like DenseNet-121 (Huang et al., 2017)
trained on radiographs (Cohen et al., 2022), we argue these adaptations remain inadequate. DenseNet-121,
despite domain alignment, represents an outdated backbone that fails to capture critical nuances, reducing
FID reliability for radiograph quality assessment. We perform an extensive ablation on this in Appendix

Section A.1. Additionally, studies using natural image-text pre-trained CLIP (Radford et al., 2021; Bluethgen et al., 2024) further compromise evaluation integrity due to high domain misalignment.

**Improving metric Reliability in CheXGenBench :**  We address the aforementioned limitations and enable a more nuanced evaluation of synthetic radiographs in CheXGenBench . For robust image fidelity assessment (FID, KID, Precision, Density, Recall, Coverage), CheXGenBench employs features from *RadDino* (Pérez-García et al., 2025), a SoTA model trained on 838K X-rays, which acts as a highly-capable feature extractor for X-rays and significantly improves metric reliability over prior works. We also report image-text alignment ("Alignment Score"), a reference-free metric functioning as a domain-specific CLIPScore (Hessel et al., 2021). We utilize *BioViL-T* (Bannur et al., 2023) to compute the global cosine similarity between the generated image and text prompt embeddings within a shared semantic space, quantifying their semantic consistency.

**Expanding the Metric Suite with Density and Mode Coverage:**  All prior studies have reported generation fidelity without a systematic evaluation of how effectively generated samples capture critical distributional characteristics, notably the density of the resulting sample distribution and coverage of distinct modes (e.g., pathologies) from the true data. This is of particular importance in medical datasets since not all pathologies are distributed equally. To address this, CheXGenBench also supports Precision, Recall, Density and Coverage (Naeem et al., 2020).

Precision, Recall, Density, and Coverage (PRDC) metrics are essential for evaluating mode coverage, particularly in long-tailed distributions where FID scores can be skewed by majority classes. This is evident in the MIMIC-CXR dataset, where "No Finding" radiographs, representing healthy X-rays, predominate despite pathological images offering greater clinical value for synthesis. Within the PRDC framework, **Precision** quantifies generated sample realism, **Recall** measures how well the real distribution is captured, **Density** assesses feature space concentration, and **Coverage** determines the proportion of modes of real data successfully generated, together providing a more comprehensive assessment than global metrics alone.

**Conditional Analysis for Individual Pathologies:** CheXGenBench extends evaluation capabilities through pathology-specific conditional analysis, wherein we compute each generation fidelity metric independently across individual pathologies in the MIMIC-CXR dataset. This granular assessment approach provides critical insights that enable developers to precisely evaluate generative performance for specific medical conditions. Such an analysis becomes particularly valuable in scenarios requiring selective augmentation of underrepresented pathologies through a generative model, which is the most desired use case. Our framework calculates FID, KID, image-text alignment, and PRDC metrics for each distinct pathology, facilitating comprehensive performance evaluation at the condition-specific level. To the best of our knowledge, we are the first study to include pathology-conditional evaluation.

**Extending CheXGenBench to Smaller Datasets:** While our empirical baselines utilize the large-scale MIMIC-CXR dataset to establish comprehensive performance standards, it is important to note that the reliability of the CheXGenBench framework is independent of the target dataset's size. Medical imaging research frequently encounters data scarcity, requiring evaluation frameworks to remain stable when applied to smaller, specialized datasets. The robustness of our benchmark stems directly from the integration of highly capable, pre-trained, domain-specific feature extractors, such as RadDino (utilized for FID and PRDC metrics) and BioViL-T (utilized for Image-Text Alignment). Because these models function as frozen feature extractors during the evaluation phase, they maintain their strong discriminative power and provide stable, reliable metrics regardless of whether they are evaluating 200,000 images or 2,000 images. Consequently, researchers applying generative architectures to low-resource domains can confidently employ CheXGenBench, assured that the underlying metric calculations remain methodologically robust.

## 2.3  Evaluation Protocol: Privacy and Patient Re-Identification

Deep generative models can inadvertently memorise distinctive training examples, allowing an attacker to reverse–engineer sensitive patient information from seemingly "synthetic" images (Carlini et al., 2023; Jegorova et al., 2023). In the medical domain, even coarse anatomical cues may be enough to link a generated

radiograph back to an individual, breaching data-protection regulations such as HIPAA[1] and the EU GDPR[2]. Consequently, to assess clinical relevance claims, a benchmark that **must** characterise (i) how much a model memorises, and (ii) how easily a real patient can be re-identified from its outputs.

**Re-identification risk formulation.** To evaluate privacy and patient re-identification risks, we implement established metrics from the existing literature (Fernandez et al., 2023; Akbar et al., 2025; Dutt et al., 2025a; Dutt, 2025). Let $\mathcal{D}_{\text{real}} = \{x_i, c_i\}_{i=1}^N$ be the training set of chest radiographs $x_i$ with corresponding captions $c_i$, and let $G_\theta$ denote a text-to-image model producing synthetic images $\hat{x} = G_\theta(c)$. We assess whether a generated sample $\hat{x}$ memorizes any training image $x_i$ via a learned similarity function $\ell(\hat{x}, x_i)$ with Memorised$(x_i) \Leftrightarrow \ell(\hat{x}, x_i) > \delta$, where $\delta$ is a fixed safety margin. We evaluate similarity through three distinct lenses: (i) **Pixel distance** ($\ell_{\text{pix}}$): $\|\hat{x} - x_i\|_2$, to detect near-duplicates (Carlini et al., 2023). (ii) **Latent distance** ($\ell_{\text{lat}}$): Normalized Euclidean distance in the embedding space of *RadDino* (Pérez-García et al., 2025). (iii) **Re-identification score** ($s_{\text{reid}}$): Probability that $\hat{x}$ and $x_i$ are from the same patient, as estimated by a Siamese neural network (Packhäuser et al., 2022).

**Assessing re-identification.** Instead of relying solely on pixel-based (Carlini et al., 2023) or structural-based (Kumar et al., 2017) similarity, we adopt the deep learning-based metric $\ell = f_\theta^{\text{re-id}}$ (Packhäuser et al., 2022). The model $f_\theta^{\text{re-id}}$ is a Siamese network with a ResNet-50 (He et al., 2016) backbone trained to classify whether two chest X-ray images originate from the same patient. For any pair $(\hat{x}, x)$ of a generated and real image, $f_\theta^{\text{re-id}}$ outputs a re-identification score $s_{\text{reid}} \in [0, 1]$ after a sigmoid layer. A synthetic image $\hat{x}$ is considered re-identified if $s_{\text{reid}} \geq \delta$ for any training image $x$. Acknowledging that any single DL-based metric can be unreliable, CheXGenBench provides a more robust privacy assessment by using pixel distance ($\ell_{\text{pix}}$) and latent distance ($\ell_{\text{lat}}$) as complementary evaluation methods.

**Formalisation of Privacy Metrics.** Given a dataset of $M$ generated images $\{\hat{x}^{(j)}\}_{j=1}^M$ (across $M$ different random seeds), let $s_{\text{reid}}^{(j)}$, $\ell_{\text{lat}}^{(j)}$, and $\ell_{\text{pix}}^{(j)}$ denote, respectively, the Re-ID score, latent-space distance, and pixel-space distance of sample $j$ to its closest training image. We report the following dataset–level statistics:

$$\textbf{Avg. Re-ID Score } (\downarrow) : \overline{s}_{\text{reid}} = \frac{1}{M}\sum_{j=1}^M s_{\text{reid}}^{(j)}, \qquad \textbf{Avg. Latent Distance } (\uparrow) : \overline{\ell}_{\text{lat}} = \frac{1}{M}\sum_{j=1}^M \ell_{\text{lat}}^{(j)},$$

$$\textbf{Avg. Pixel Distance } (\uparrow) : \overline{\ell}_{\text{pix}} = \frac{1}{M}\sum_{j=1}^M \ell_{\text{pix}}^{(j)}, \qquad \textbf{Max. Re-ID Score } (\downarrow) : s_{\text{reid}}^{\text{max}} = \max_{1 \leq j \leq M} s_{\text{reid}}^{(j)},$$

$$\textbf{Count}[s_{\text{reid}} > \delta] (\downarrow) : C_\delta = \sum_{j=1}^M \mathbf{1}\big[s_{\text{reid}}^{(j)} > \delta\big].$$

## 2.4 Evaluation Protocol: Synthetic Data Utility for Downstream Tasks

A significant application of synthetic data in radiology lies in enhancing downstream model performance, potentially circumventing the stringent sharing restrictions imposed on medical datasets. For our utility assessment framework, we have strategically selected three prevalent downstream tasks in medical image analysis: **(1) Image Classification**, **(2) Image Segmentation**, and **(3) Radiology Report Generation (RRG)**. Classification and Segmentation serve as unimodal tasks, assessing the synthetic images across two dimensions: texture and topology. RRG functions as a more demanding multimodal evaluation, assessing the factual correctness between synthetic images and their corresponding clinical descriptions. Collectively, these tasks present distinct complexities for a comprehensive assessment of the utility of synthetic data.

**Downstream Image Classification:** We assess the complementary value of synthetic data when integrated with real-world clinical datasets. Specifically, we train a classifier (ResNet50 (He et al., 2016)) on a combination of $\mathcal{D}_{real}$ and $\mathcal{D}_{syn}$ (20K + 20K samples) for 20 epochs. This 20-epoch training schedule was determined empirically. During initial experiments, we observed that validation metrics consistently saturated by the 20th

---

[1]https://www.hhs.gov/hipaa/for-professionals/privacy/index.htm
[2]http://gdpr.eu/what-is-gdpr/

epoch, with further training merely introducing the risk of overfitting to the synthetic data augmentations. The performance metrics are calculated on a held-out real test set ($\mathcal{D}_{test}$, consisting of exactly 5,034 samples corresponding to the official MIMIC-CXR test split,) to ensure clinical relevance and generalizability of our findings. We quantify classification performance through standard accuracy, F1-Score, and AUROC metrics.

**Downstream Image Segmentation:** The MIMIC-CXR dataset lacks pixel-level annotations, hence, we established a surrogate task of segmenting a specific anatomical structure, the *clavicles*. First, we generated *pseudo* ground-truth masks for clavicles using the medical segmentation model proposed by Seibold et al. (2023), which is capable of delineating 159 distinct anatomical regions. To assess the complementary benefits of synthetic data, we supplemented 3000 real image-mask pairs ($\mathcal{D}_{real}$) with 3000 synthetic pairs ($\mathcal{D}_{syn}$) from each T2I model in our evaluation. We trained a U-Net (Ronneberger et al., 2015) using this data and report the results on the held-out test set ($\mathcal{D}_{test}$).

**Downstream Report Generation:** To study the impact of synthetic data augmentation for RRG, we designed a simplified training protocol. We constructed a baseline training set using a stratified subset of 50K real samples ($\mathcal{D}_{real}$) from MIMIC-CXR, preserving the original prevalence ratios of all pathology classes. Next, we supplemented this baseline with 20K synthetic samples generated by each candidate T2I model ($\mathcal{D}_{syn}$) and train *LLaVA-Rad* (Zambrano Chaves et al., 2025) from scratch. We quantify the performance on $\mathcal{D}_{test}$ using the BLEU (Papineni et al., 2002), ROUGE-L (Lin, 2004), F1-RadGraph (Jain et al., 2021), GREENScore (Ostmeier et al., 2024) and and RaTEScore (Zhao et al., 2024) metrics.

**Quantifying the Exclusive Utility of Synthetic Data:** Beyond investigating the utility of synthetic data in complementing real data, we evaluated its *exclusive* utility by performing downstream tasks using synthetic datasets in isolation. This analysis focused specifically on Classification and Report Generation. Detailed descriptions of the experimental framework and the corresponding results are documented in Appendix A.7.

## 3  Experiments and Results

**Training Setting (T2I Training):** Our evaluation framework explicitly comprises both major foundation models from the natural imaging domain fine-tuned on the MIMIC-CXR dataset (Johnson et al., 2016; 2019), as well as existing domain-specific models tailored for radiograph generation. For the domain-specific baselines, we evaluated existing radiograph generation models (RadEdit (Pérez-García et al., 2024), LLM-CXR (Lee et al., 2024)) out-of-the-box without further modification. For all other foundation architectures, the fine-tuning strategy was determined by model scale. Models with fewer than 1B parameters underwent full fine-tuning (FFT), whereas larger models (>1B parameters) utilized Parameter-Efficient Fine-Tuning (PEFT) via LoRA (Hu et al., 2022) with a rank of 32 applied to the query, key, and value layers. The training hyperparameters are presented in Appendix A.5. Furthermore, we conduct additional ablations on the rank and position of LoRA in Appendix A.6.

Implementation frameworks varied by model family. For the Stable-Diffusion variants, we adopted their implementation and training protocols from Huggingface Diffusers (von Platen et al., 2022), while Sana and Pixart-Sigma were implemented via their official repositories. High-capacity models were managed using the `ai-toolkit` package[3]. All training sessions adhered to officially recommended hyperparameters, maintaining a consistent total batch size of 128 across four Nvidia H200 GPUs. The core dataset utilized for both fine-tuning and downstream evaluation is the widely adopted, publicly available MIMIC-CXR dataset, paired specifically with the recently released LLaVA-Rad enhanced annotations (Zambrano Chaves et al., 2025). MIMIC-CXR consists of chest X-rays paired with corresponding clinical text and pathological annotations, serving as a standard for text-conditional radiograph generation. To ensure a strictly fair comparison across all evaluated architectures, we utilized an identical training split comprising 237,388 samples and a held-out test split of 5,034 samples. Regarding our data release, we are publicly releasing our standardized evaluation splits, the corresponding enhanced annotations, and the generated synthetic image outputs from all baseline models to facilitate seamless reproducibility and future comparative studies. Subsequent downstream evaluations were performed on Nvidia A100 GPUs.

---

[3]https://github.com/ostris/ai-toolkit

Table 1: Table comparing the results for generative fidelity for 11 different T2I models in the benchmark. The best result for each metric is indicated with **bold**, while the second-best result is underlined. The overall best performing model is  highlighted in green .

| Model | Size | Default Resolution | Prev. Available For CXR? | Fine-Tuning | FID ↓ (RadDino) | KID ↓ (RadDino) | Alignment Score ↑ | Precision ↑ | Recall ↑ | Density ↑ | Coverage ↑ |
|---|---|---|---|---|---|---|---|---|---|---|---|
| **SD V1-4** (Rombach et al., 2022) | 0.86B | 512 | ✓ | FFT | 125.186 | 0.172 | 0.357 | 0.488 | 0.301 | 0.236 | 0.217 |
| **SD V1-5** (Rombach et al., 2022) | 0.86B | 512 | ✓ | FFT | 118.932 | 0.147 | 0.326 | 0.536 | 0.473 | 0.242 | 0.256 |
| **SD V2** (Rombach et al., 2022) | 0.86B | 512 | ✓ | FFT | 194.724 | 0.376 | 0.311 | 0.480 | 0.086 | 0.166 | 0.057 |
| **SD V2-1** (Rombach et al., 2022) | 0.86B | 512 | ✓ | FFT | 186.530 | 0.413 | 0.197 | 0.530 | 0.049 | 0.180 | 0.038 |
| **RadEdit** (Pérez-García et al., 2024) | 0.86B | 512 | ✓ | N/A | 69.695 | 0.033 | 0.677 | 0.397 | 0.544 | 0.150 | 0.285 |
| **Pixart Sigma** (Chen et al., 2024) | 0.60B | 512 | ✗ | FFT | 60.154 | 0.023 | **0.697** | 0.666 | 0.522 | 0.506 | 0.506 |
| **Sana** (Xie et al., 2025) | 0.60B | 512 | ✗ | FFT | **54.225** | **0.016** | 0.695 | 0.674 | **0.614** | 0.520 | **0.548** |
| **SD V3.5 Medium** (Esser et al., 2024) | 2.50B | 1024 | ✗ | LoRA(r=32) | 91.302 | 0.103 | 0.044 | 0.632 | 0.205 | 0.401 | 0.244 |
| **Lumina 2.0** (Qin et al., 2025) | 2.60B | 1024 | ✗ | LoRA(r=32) | 101.198 | 0.110 | 0.121 | 0.574 | 0.014 | 0.256 | 0.170 |
| **Flux.1-Dev** (Labs, 2024) | 12B | 1024 | ✗ | LoRA(r=32) | 122.400 | 0.144 | 0.036 | 0.420 | 0.008 | 0.125 | 0.326 |
| **LLM-CXR** (Lee et al., 2024) | 12B | 256 | ✓ | N/A | 71.243 | 0.061 | 0.319 | **0.782** | 0.041 | **0.671** | 0.459 |

## 3.1 Fidelity of Synthetic Generations

### Results on Global Assessment

The results are presented in Tab. 1, where we showcase both fidelity and mode coverage metrics. Sana (Xie et al., 2025) delivers superior overall performance across key metrics, achieving the lowest FID and KID scores, indicating exceptional generation fidelity, while simultaneously attaining the highest Recall and Coverage, demonstrating its capacity to capture diverse modes (distributions) throughout the dataset. Pixart Sigma (Chen et al., 2024) emerges as a strong contender, exhibiting the highest image-text alignment alongside second-best FID, KID, and Coverage scores. To better understand mode coverage, we conducted a correlation analysis (detailed in Appendix A.3) between pathology prevalence in the training data and generative fidelity (FID). We observed a remarkably strong positive correlation coefficient of 0.947, demonstrating a direct relationship between a disease's frequency in the training data and the model's ability to accurately generate it, confirming that models exhibit severe bias toward dense data regions. LLM-CXR (Lee et al., 2024) exhibits specialised capabilities, substantially outperforming all other models in Precision; however, its notably low Recall suggests limited generative scope, primarily effective for specific distributions (pathologies). This also highlights that conventional fidelity metrics like FID do not present a complete picture of the model performance. Earlier Stable-Diffusion variants (SD V1-x, V2-x) demonstrate consistently suboptimal scores across all metrics despite full fine-tuning, a particularly significant finding given their prevalent adoption in the synthetic radiograph generation literature (Bluethgen et al., 2024; Favero et al., 2025; Fernandez et al., 2023; Dutt et al., 2024b). Larger architectural models (SD V3-5 (Esser et al., 2024), Lumina 2.0 (Qin et al., 2025), Flux.1-Dev (Labs, 2024)), with the exception of LLM-CXR, yield predominantly inferior performance across evaluation metrics. SD V3-5 exhibits high Precision but low Recall, Density, and Coverage scores, mirroring trends observed in LLM-CXR. We hypothesise that this stems from the inability of LoRA to provide sufficient adaptation for the medical domain, a limitation previously observed in Dutt et al. (2024b). Performance improvements might be achievable by extending LoRA to other linear layers beyond attention (Q,K,V) layers, however, we leave this exploration to future work. Overall, Sana achieves the optimal performance-efficiency trade-off among all evaluated models. **Notably**, Sana has been adapted for synthetic radiograph generation for the first time through this work.

### Results on Conditional Assessment

Results for stratified analysis over individual pathologies are presented in Tab. 2. Consistent with trends observed in Tab. 1, Sana demonstrates superior performance, achieving the lowest FID scores across 12 of the 14 pathology categories. This indicates Sana's robust capability to generate high-fidelity radiographs across diverse pathological conditions. Pixart Sigma maintains its position as the second-best performing model, while RadEdit frequently secures the third-best scores across multiple categories. LLM-CXR demonstrates competitive performance for specific pathologies, notably achieving strong results for Edema (83.18) and No Finding (64.62), frequently outperforming both earlier Stable Diffusion models and certain large-scale models

---

[1] **Note:** **EC** = Enlarged Cardiomediastinum, **LL** = Lung Lesion, **LO** = Lung Opacity, **NF** = No Finding, **PE** = Pleural Effusion, **PO** = Pleural Other, **PN** = Pneumonia, **PT** = Pneumothorax, **SD** = Support Devices.

Table 2: Comparison of FID (RadDino) scores (↓) across individual pathologies in the MIMIC-CXR dataset. Lower values indicate superior performance, with the best results for each pathology highlighted in **bold**. The most challenging pathology (highest average FID across models) is highlighted in red , while the best-performing model overall is highlighted in green . We also highlight the best performing pathology for each model in blue .

| Model | Atelectasis | Cardiomegaly | Consolid. | Edema | EC | Fracture | LL | LO | NF | PE | PO | PN | PT | SD |
|---|---|---|---|---|---|---|---|---|---|---|---|---|---|---|
| SD V1-4 | 134.11 | 131.04 | 184.30 | 144.84 | 217.75 | 238.78 | 225.99 | 129.38 | 106.34 | 128.16 | 255.84 | 163.82 | 212.48 | 135.10 |
| SD V1-5 | 125.67 | 124.75 | 181.25 | 139.94 | 213.94 | 243.17 | 123.13 | 255.64 | 167.81 | 193.75 | 243.17 | 101.08 | 119.91 | 123.64 |
| SD V2 | 188.72 | 193.91 | 241.24 | 214.40 | 214.40 | 253.91 | 268.28 | 280.11 | 193.99 | 299.48 | 223.43 | 250.96 | 183.34 | 193.99 |
| SD V2-1 | 179.20 | 181.79 | 228.43 | 193.62 | 242.65 | 263.01 | 260.15 | 185.00 | 192.30 | 178.84 | 287.27 | 213.26 | 242.60 | 176.99 |
| RadEdit | 63.38 | 62.79 | 136.59 | 76.94 | 155.97 | 197.58 | 184.11 | 61.90 | 67.88 | 60.60 | 215.92 | 114.66 | 151.34 | 53.10 |
| Pixart Sigma | 59.27 | 60.39 | 133.96 | 73.93 | 155.53 | 179.44 | 174.63 | 56.83 | 48.74 | 59.05 | 210.90 | 108.42 | 150.55 | 51.61 |
| Sana | 51.03 | 54.68 | 127.46 | 67.84 | 147.00 | 172.32 | 163.14 | 49.23 | 44.60 | 49.80 | 199.45 | 99.52 | 141.99 | 46.51 |
| SD V3.5 Medium | 94.94 | 94.84 | 149.05 | 111.94 | 168.48 | 184.75 | 173.37 | 86.72 | 89.60 | 91.92 | 203.62 | 124.07 | 163.27 | 86.99 |
| Lumina 2.0 | 109.39 | 111.11 | 162.36 | 131.18 | 182.35 | 191.83 | 182.22 | 99.53 | 95.66 | 105.25 | 213.50 | 134.58 | 165.09 | 102.78 |
| Flux.1-Dev | 137.10 | 133.60 | 176.76 | 152.91 | 191.48 | 191.02 | 194.97 | 133.37 | 100.58 | 137.66 | 221.23 | 156.59 | 190.93 | 127.03 |
| LLM-CXR | 71.57 | 71.37 | 136.65 | 83.18 | 148.28 | 168.50 | 163.22 | 66.93 | 64.62 | 67.83 | 200.84 | 108.04 | 147.52 | 67.54 |

Table 3: Results on Re-Identification Risk and Patient Privacy Metrics. We present the average scores across 2000 samples and individual scores with maximum privacy risks.

| Model | SD V1-4 | SD V1-5 | SD V2 | SD V2-1 | RadEdit | Sana | Pixart-Σ | SD V3-5 | Lum. 2.0 | Flux | LLM-CXR |
|---|---|---|---|---|---|---|---|---|---|---|---|
| Avg. Re-ID Score (↓) | 0.539 | 0.572 | 0.533 | 0.503 | 0.481 | 0.551 | 0.548 | **0.365** | 0.513 | 0.404 | 0.537 |
| Avg. Latent Distance (↑) | 0.592 | 0.583 | 0.588 | 0.592 | 0.560 | 0.540 | 0.561 | **0.601** | 0.591 | 0.595 | 0.557 |
| Avg. Pixel Distance (↑) | 143 | 143 | 143 | 145 | 145 | **162** | 159 | 147 | 145 | 155 | 149 |
| Max. Re-ID Score (↓) | 0.996 | 0.996 | 0.996 | 0.997 | 0.992 | 0.996 | 0.994 | 0.997 | 0.993 | 0.992 | 0.994 |
| Count Re-ID > δ (↓) | 434 | 498 | 454 | 392 | 380 | 442 | 442 | 236 | 223 | 196 | 419 |

(SD V3.5, Lumina, Flux.1-Dev).

**Concerning Observations:** This analysis reveals concerning patterns. Substantial performance variation exists across pathologies for each model, regardless of overall performance. For instance, Sana exhibits FID scores ranging from 44.60 for "*No Finding (NF)*" to 199.45 for "*Pleural Other (PO)*". Notably, five of the eleven models achieve optimal performance on "No Finding" cases, which represent healthy radiographs with limited clinical utility from synthetic X-rays, while all models consistently perform poorly on "*Pleural Other (PO)*" pathology. In Appendix A.3 and Tab. 9, we empirically demonstrate that model performance strongly correlates with pathology prevalence in the training dataset (**correlation coefficient: 0.947**), suggesting that current models mainly reproduce the largest modes in the dataset, while failing to model the longer tail of pathologies, and thus fail to achieve general clinical utility. Thus future medical image generation models should follow this evaluation protocol in order to make claims of clinical utility. We also hope this analysis encourages developers to incorporate training strategies tailored for long-tailed distributions (Qin et al., 2023).

## 3.2 Results on Privacy and Re-Identification Risks

**Experimental Setting:** To quantify re-identification risks, we select a subset of 2000 image-text pairs $(x_i^{img}, x_i^{txt})$ from the training set and generate $N(=10)$ synthetic samples $\hat{x}_i^{img,1}, \hat{x}_i^{img,2}, \ldots, \hat{x}_i^{img,N}$ using 10 different random seeds for each training prompt. Next, we calculate Re-ID scores, Pixel and Latent Distances between each real-synthetic pair $(x_i^{img}, \hat{x}_i^{img,n})$ for all $n \in \{1, 2, \ldots, N\}$. Finally, we report the maximum Re-ID score $\max_j s_{\text{reid}}^{(j)}$, minimum pixel distance $\min_j \ell_{\text{pix}}^{(j)}$ and minimum latent distance $\min_j \ell_{\text{lat}}^{(j)}$ across $N$ generations for each sample. This approach enables us to identify the greatest privacy risk posed for each training sample across multiple generations.

**Results:** The privacy risk assessment results are detailed in Tab. 3. Most models exhibit Average Re-ID scores within a comparable range, with SD V3-5 notably achieving the lowest (most favourable) score. For latent and pixel distances, a similar pattern emerges, where SD V3-5 and Sana demonstrate superior performance (i.e., lower average distances), respectively. **Concerns:** We conducted a detailed analysis of *individual* scores across 2,000 samples, with particular attention to two key metrics: (1) the maximum Re-ID

Table 4: Comparison of baseline (Real Data) against Synthetic Augmentation. The final column represents the row-wise average AUC. Instances where performance gain for minority classes (Fracture, PO) is observed are  highlighted .

| Model | Atelec. | C.Megaly | Consol. | Edema | Enlarged C. | Frac. | LL | LO | NF | PE | PO | PN | PT | SD | Average |
|---|---|---|---|---|---|---|---|---|---|---|---|---|---|---|---|
| Original | 0.75 | 0.76 | 0.72 | 0.85 | 0.61 | 0.58 | 0.63 | 0.70 | 0.84 | 0.84 | 0.74 | 0.67 | 0.71 | 0.83 | 0.73 |
| SD V1-4 | 0.76 | 0.78 | 0.74 | 0.87 | 0.65 | 0.60 | 0.73 | 0.73 | 0.85 | 0.85 | 0.74 | 0.70 | 0.77 | 0.87 | **0.76** |
| SD V1-5 | 0.76 | 0.77 | 0.74 | 0.88 | 0.66 | 0.60 | 0.72 | 0.73 | 0.85 | 0.86 | 0.72 | 0.71 | 0.77 | 0.87 | **0.76** |
| SD V2 | 0.76 | 0.78 | 0.73 | 0.87 | 0.64 | 0.58 | 0.69 | 0.72 | 0.85 | 0.86 | 0.70 | 0.69 | 0.76 | 0.86 | **0.75** |
| SD V2-1 | 0.76 | 0.77 | 0.73 | 0.87 | 0.64 | 0.58 | 0.68 | 0.72 | 0.85 | 0.85 | 0.72 | 0.69 | 0.75 | 0.86 | **0.75** |
| RadEdit | 0.76 | 0.77 | 0.75 | 0.87 | 0.66 | 0.62 | 0.70 | 0.72 | 0.85 | 0.86 | 0.78 | 0.69 | 0.77 | 0.85 | **0.76** |
| Pixart $\Sigma$ | 0.76 | 0.78 | 0.75 | 0.87 | 0.66 | 0.62 | 0.73 | 0.73 | 0.85 | 0.85 | 0.75 | 0.70 | 0.78 | 0.86 | **0.76** |
| Sana | 0.76 | 0.78 | 0.74 | 0.87 | 0.66 | 0.63 | 0.70 | 0.73 | 0.85 | 0.86 | 0.77 | 0.69 | 0.78 | 0.86 | **0.76** |
| SD V3-5 | 0.74 | 0.76 | 0.72 | 0.85 | 0.63 | 0.61 | 0.65 | 0.70 | 0.84 | 0.84 | 0.70 | 0.66 | 0.72 | 0.84 | 0.73 |
| Lumina 2.0 | 0.74 | 0.74 | 0.73 | 0.85 | 0.63 | 0.62 | 0.65 | 0.71 | 0.84 | 0.84 | 0.65 | 0.65 | 0.70 | 0.84 | 0.73 |
| Flux.1-Dev | 0.75 | 0.72 | 0.72 | 0.84 | 0.61 | 0.61 | 0.66 | 0.69 | 0.84 | 0.83 | 0.69 | 0.66 | 0.68 | 0.83 | 0.73 |
| LLM-CXR | 0.76 | 0.76 | 0.74 | 0.87 | 0.66 | 0.59 | 0.69 | 0.70 | 0.82 | 0.83 | 0.74 | 0.68 | 0.75 | 0.85 | **0.75** |

score, which represents the highest potential for re-identification, and (2) the frequency of samples exceeding a high-risk threshold ($\delta = 0.85$). Our results reveal that all models, regardless of their fidelity performance, generate samples that can be re-identified with high confidence. The proportion of samples presenting significant re-identification risk remains substantial across all models, ranging from 10% to 25%. Notably, models trained with LoRA demonstrate a relatively lower incidence of high-risk samples, potentially due to their reduced capacity for memorization (Dutt et al., 2025a). These findings underscore a critical insight: **generative models pose substantial privacy risks irrespective of their generative capabilities.**

**Real-Data Baseline and Stratified Privacy Risks:** To provide physical context for these metrics and establish a realistic limit for patient re-identification, we computed a real-data baseline. Specifically, we evaluated the Siamese Re-ID network on actual longitudinal pairs drawn from the MIMIC-CXR dataset. We found that the average Re-ID score on real patient data is 0.59. Crucially, the average re-identification risk from our synthetic data almost perfectly matches that of this real-data baseline. As shown in Tab. 3, the average re-identification scores for models like Sana (0.551) and SD V1-5 (0.572) lie very close to 0.59, demonstrating that an average synthetic image carries a similar re-identification risk as an authentic X-ray scan. The primary motivation for training generative models in healthcare is to circumvent privacy restrictions. However, our results indicate that current generative architectures provide a false sense of anonymization, critically undermining the safety of openly sharing synthetic medical datasets.

**Stratified Privacy Risks:** The risks mentioned above are further compounded when stratifying our analysis based on the presence of pathological findings. We discovered that diseased synthetic radiographs exhibit notably higher re-identification risks compared to "healthy" synthetic radiographs. This indicates that models are more prone to memorizing patient-specific anatomical details when conditioned on distinct pathological features. We highlight this as a critical warning for the field. Privacy evaluations must be inherently pathology-aware, as assessing privacy risk purely on aggregate distributions will systematically underestimate the vulnerability of the most clinically sensitive patient cohorts.

## 3.3 Utility of Synthetic Samples for Downstream Tasks

**Downstream Image Classification:** The results presented in Tab. 4, show an interesting phenomenon. On average, data from each T2I model (irrespective of the generative fidelity) either matches or outperforms the real data baseline. For instance, SD V1-4 (FID $\approx$ 125) performs equally well as Sana (FID $\approx$ 54), leading to a 3% improvement in the average AUC. Furthermore, minority pathologies such as *Pleural Other*, where most T2I models demonstrated low FID scores, outperform the real data baseline on several instances (RadEdit, Pixart-$\Sigma$, Sana). For most prevalent pathologies such as Atelectasis (AT), Cardiomegaly (CM), and Support Devices (SD), most T2I models either match or outperform the real data baseline. Overall, the results demonstrate that synthetic data augmentation consistently provides complementary value for downstream image classification, largely independent of the source model's generative fidelity. Even lower-fidelity models yield performance gains comparable to state-of-the-art architectures, effectively matching

or exceeding the real-data baseline. Notably, this augmentation proves particularly beneficial for minority pathologies, suggesting that current T2I models successfully capture and preserve the macroscopic topological features necessary for classification tasks, even when microscopic texture is lacking. We discuss this further in Sec. 4.

Table 5: Comparing the utility of synthetic data augmentation across T2I models for image segmentation.

| Augmentation Source | DICE (n=6,000) | DICE (n=10,000) |
|---|---|---|
| **Baseline (n=3,000)** | 0.669 | |
| *Augmented Data* | | |
| SD V1-4 | 0.593 | 0.575 |
| SD V1-5 | 0.586 | 0.566 |
| SD V2 | 0.537 | 0.512 |
| SD V2-1 | 0.541 | 0.537 |
| RadEdit | 0.667 | 0.671 |
| Pixart Σ | **0.671** | **0.679** |
| Sana | 0.667 | **0.679** |
| SD V3-5 | 0.612 | 0.601 |
| Lumina 2 | 0.581 | 0.556 |
| Flux.1 | 0.562 | 0.531 |
| LLM-CXR | 0.642 | 0.633 |

**Downstream Image Segmentation:** The results for downstream image segmentation, presented in Table 5, reveal that the utility of synthetic data for pixel-level tasks is highly dependent on the generative fidelity of the source text-to-image model. Unlike image classification, where even lower-quality synthetic data provided benefits, augmenting the baseline 3,000 real images (DICE score of 0.669) with data from low-fidelity models such as SD V1-4, SD V1-5, and Lumina 2.0 led to substantial performance degradation. For instance, supplementing the training set with SD-V2 generations resulted in a sharp DICE score drop to 0.537. Conversely, high-fidelity models like Pixart Sigma, Sana, and RadEdit were able to preserve necessary structural integrity, successfully matching or marginally surpassing the real-data baseline (achieving a peak augmented DICE of 0.671 with Pixart Sigma). This distinct contrast underscores that state-of-the-art generative models are required to yield meaningful benefits for precise, texture-sensitive medical image analysis tasks.

**Downstream Radiology Report Generation:** As detailed in Table 6, the results for the downstream radiology report generation (RRG) task showcase a strong dependency on the generative fidelity of the synthetic data. In contrast to the classification results, augmenting the baseline dataset of 50,000 real images with 20,000 samples from low-fidelity models (such as SD V1-4 and SD V1-5) led to a notable degradation across most evaluation metrics. For example, integrating SD V1-4 generations caused the BLEU-4 score to fall from the real-data baseline of 10.29 to 8.76, and the F1-RadGraph score to decrease from 0.21 to 0.19. Conversely, supplementing the training set with images from high-fidelity models like RadEdit, Pixart Sigma, and Sana yielded substantial performance improvements. Sana, for instance, boosted the BLEU-4 score to 11.28 and the F1-RadGraph score to 0.26. We hypothesize that because RRG is a highly texture-sensitive multimodal task, the "waxy" or denoised artefacts present in low-fidelity generations likely *confuse* the visual encoder. This confusion induces hallucinations or generic descriptions that fail to align with granular clinical ground truths, whereas high-quality generations successfully preserve the essential microscopic texture required for accurate performance.

## 4 Synthetic Data Semantics: The Texture-Topology Gap

In this section, we dive deeper into the semantics of the synthetic images and study how they compare against the real images. This analysis would enable us to understand *where* and *why* synthetic data augmentation provides performance gains.

Table 6: Performance comparison of LLaVA-Rad trained from scratch with real data only versus real data augmented with synthetic images from various text-to-image models ( low-fidelity and high-fidelity ).

| Metric | Original (Real Data Only) (N=50,000) | Real Data + SD V1-4 (N=70,000) | Real Data + SD V1-5 (N=70,000) | Real Data + RadEdit (N=70,000) | Real Data + Pixart (N=70,000) | Real Data + Sana (N=70,000) |
|---|---|---|---|---|---|---|
| BLEU-1 (↑) | 27.26 | 24.49 | 25.62 | 31.40 | 32.53 | 32.83 |
| BLEU-4 (↑) | 10.29 | 8.76 | 8.95 | 10.49 | 11.26 | 11.28 |
| ROUGE-L (↑) | 0.23 | 0.20 | 0.22 | 0.26 | 0.26 | 0.26 |
| F1-RadGraph (↑) | 0.21 | 0.19 | 0.22 | 0.26 | 0.26 | 0.26 |
| Micro F1-5 (↑) | 0.45 | 0.41 | 0.41 | 0.49 | 0.51 | 0.52 |
| Micro F1-14 (↑) | 0.45 | 0.39 | 0.40 | 0.50 | 0.52 | 0.52 |
| GREENScore (↑) | 0.27 | 0.27 | 0.27 | 0.31 | 0.31 | 0.32 |
| RaTEScore (↑) | 0.45 | 0.49 | 0.49 | 0.50 | 0.52 | 0.53 |

## 4.1 Explaining the Results with the Texture-Topology Gap

We observe distinct trends in the utility of synthetic data across classification, segmentation, and radiology report generation (RRG) tasks. While classification tasks benefit from both low- and high-fidelity synthetic data, segmentation and RRG demonstrate improvements only with high-fidelity data. We attribute this phenomenon to the Texture-Topology Gap.

**Macroscopic Topology vs. Microscopic Texture:** To understand this divergence in performance across tasks, it is necessary to decouple the visual components of medical images. We define two core components: **macroscopic topology** and **microscopic texture**. The macroscopic topology of a radiograph represents the geometric arrangement and spatial relationships of the different anatomical structures. This includes the shape of the cardiac silhouette, the curvature of the rib cage, and the presence of opacities (often indicating pathologies). On the other hand, the microscopic details include high-frequency, fine-grained features such as bone granularity, intricate pulmonary vascular branching patterns, and the inherent signal-dependent noise in the X-Ray due to data acquisition protocols. We provide a qualitative comparison between real and synthetic radiographs in Fig. 2.

**Task-Specific Impacts of the Domain Shift:** Visual inspection reveals that this macroscopic topology is consistently preserved in both high and low-fidelity images, i.e. on a global level, the structure, shape and position of the different anatomical components is preserved. In image classification, networks predominantly rely on the global structural cues and the presence or absence of large-scale anomalies, this preserved topology translates directly to performance gains in tasks like image classification, effectively serving as robust regularization.

However, synthetic generations consistently fail to reproduce the crucial microscopic details. For instance, in Fig. 2, we can observe that the synthetic radiographs, especially ones with low-fidelity (bottom row), exhibit a smoothed, "waxy" appearance of the bones. This distinction creates a severe domain shift. Low-fidelity data degrades the performance for texture-sensitive tasks like segmentation and report generation because these tasks require dense, pixel-level representations. In segmentation, the "waxy" smoothing degrades the sharp local gradients required to delineate precise anatomical boundaries. Similarly, in RRG, the absence of high-frequency microscopic texture deprives the multimodal vision encoder of the granular evidence required to ground complex clinical terminology, leading to hallucinations and generic text generation.

**An Open Algorithmic Challenge:** Based on these findings, we explicitly frame the Texture-Topology Gap not merely as an observational limitation, but as a primary open algorithmic challenge for the field of generative medical AI. Closing this gap requires the development of novel generative objectives, sampling strategies, and architectural innovations that can simultaneously model macroscopic anatomical coherence and high-frequency microscopic fidelity. In this context, CheXGenBench is designed to serve as a rigorous

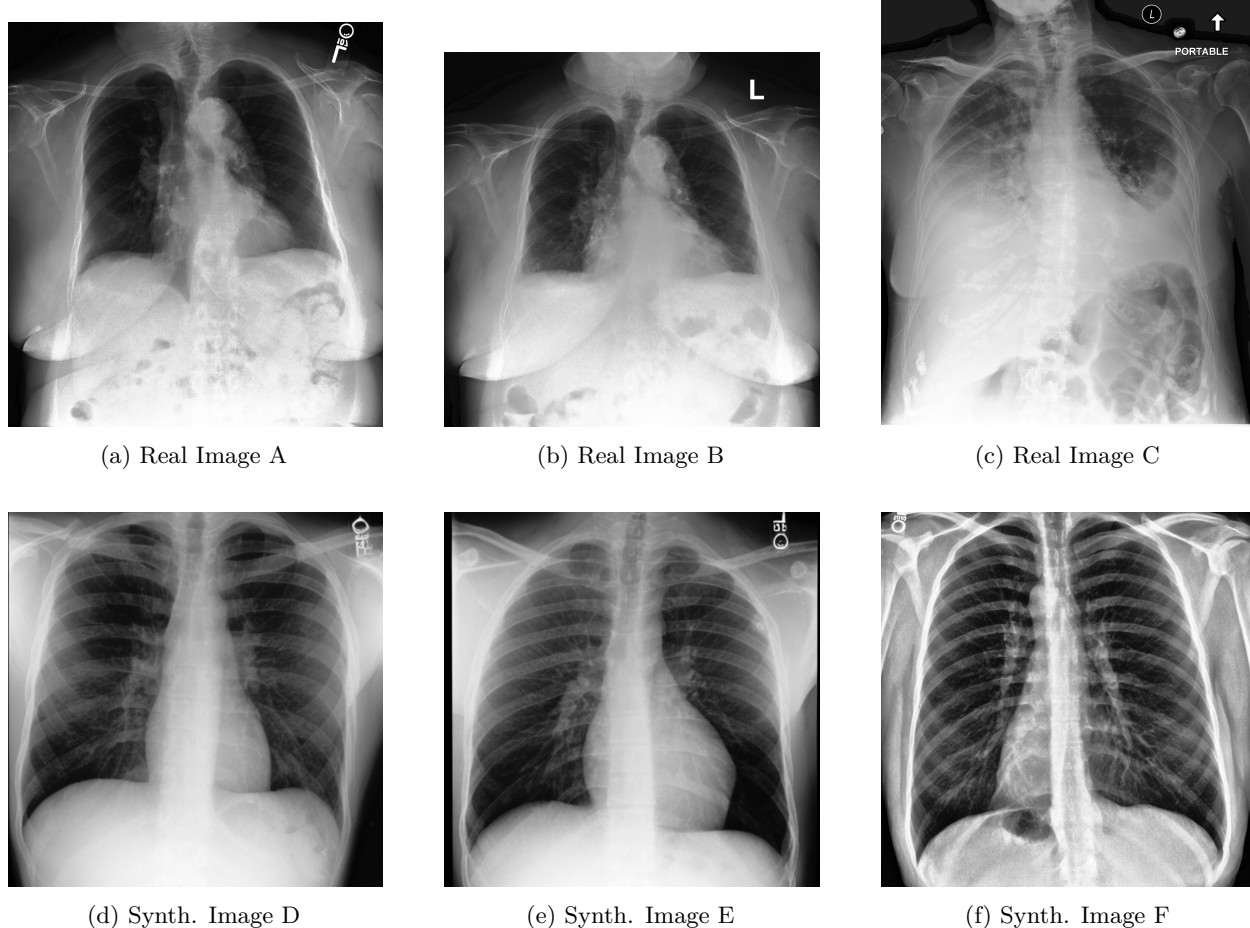

(a) Real Image A       (b) Real Image B       (c) Real Image C

(d) Synth. Image D       (e) Synth. Image E       (f) Synth. Image F

Figure 2: Comparison of texture and appearance between real and synthetic radiographs. **Top row:** Real data; **Bottom row:** Synthetic data.

diagnostic tool, providing the standardized multi-task utility metrics necessary to quantify this gap and track algorithmic progress toward clinically realistic synthetic generation.

## 5 Conclusion

We've addressed critical gaps in synthetic radiograph generation research by introducing CheXGenBench, a unified framework to assess generation fidelity, privacy, and clinical utility. Our work highlights key limitations of current models: even state-of-the-art (SoTA) models struggle with long-tailed data distributions, and those with poor fidelity can still pose significant privacy risks. Additionally, while synthetic data shows promise for unimodal tasks, its utility for more complex, multimodal applications remains limited. These observations provide key research directions for future work in synthetic radiograph generation. We envision CheXGenBench to grow with new generative models and paradigms and serve as a dynamic standard for the medical AI community.

In summary, this work contributes (i) a unified, extensible benchmark that couples global and pathology-conditional fidelity and mode-coverage evaluation with rigorous privacy assessment, and (ii) a practical utility suite spanning classification, segmentation, and radiology report generation. Across 11 modern T2I systems, we observe that the strongest models achieve substantial gains in fidelity and coverage, yet still exhibit pronounced weaknesses on rare pathologies, underscoring the need for long-tail aware training and evaluation.

Our results also indicate that privacy risks persist across model families and are not reliably predicted by fidelity alone. We therefore recommend that future work treat privacy evaluation as a first-class criterion (alongside fidelity and utility), and that synthetic data be shared and deployed with explicit risk characterization, access controls, and clear intended-use limitations.

Finally, CheXGenBench opens several promising directions: (1) developing generative objectives and sampling strategies that better model rare pathologies and clinically meaningful variation, (2) designing privacy-preserving fine-tuning and generation mechanisms with verifiable guarantees, and (3) closing the texture-topology gap to improve synthetic utility for pixel-level and multimodal tasks. While our current downstream evaluations establish a foundational baseline, a critical next step is expanding these utility metrics to include more complex, granular diagnostic tasks. Specifically, integrating precise pathology localization (e.g., via bounding boxes), medical Visual Question Answering (VQA), and advanced open-ended radiological report generation will provide a more rigorous clinical stress test of these models. We believe that standardised benchmarking through CheXGenBench will accelerate progress toward high-fidelity, privacy-aware, and clinically useful synthetic radiographs.

### 5.1 Limitations

**Data Contamination and Foundation Models:** A potential limitation when evaluating foundation models is the inherent risk of data contamination during their initial pre-training phase. Architectures such as Stable Diffusion, Flux, and Lumina are pre-trained on massive, scraped web corpora (e.g., LAION-5B), raising the theoretical possibility that they may have ingested general medical imagery. However, we assess the risk of direct contamination with our target dataset to be highly improbable due to two key factors. First, the MIMIC-CXR dataset is hosted behind strict institutional firewalls on the PhysioNet platform and requires a signed, credentialed Data Use Agreement for access, effectively shielding it from standard large-scale internet scrapers. Second, this structural protection is corroborated by empirical evidence from existing literature, which consistently demonstrates that general-purpose models exhibit poor zero-shot generalization to the highly specialized medical domain. If these models had ingested the MIMIC-CXR dataset during pre-training, one would expect significantly higher zero-shot clinical fidelity out-of-the-box. Consequently, the combination of gated data access and established domain shift strongly suggests an absence of direct data contamination in the base models evaluated in this benchmark.

## 6 Broader Impact Statement

While synthetic medical data is frequently viewed as a privacy-preserving alternative to real patient data, our CheXGenBench benchmark demonstrates that this is often a false guarantee. Current text-to-image models can essentially act as complex compression algorithms that can memorize sensitive biometric features. Because the re-identification risk of the generated images in our study closely mirrors the risk of real radiographs, openly sharing synthetic datasets or model weights carries severe privacy implications. Crucially, we also found that patients with rare or severe pathologies are significantly more vulnerable to data memorization than healthy individuals.

This vulnerability is tied to a broader failure to generalize across the long tail distribution of medical data. Generative models tend to prioritize common, dense data regions and collapse when modeling underrepresented conditions (rare pathologies). If downstream clinical decision systems are trained using this biased synthetic data, there is a risk of reinforcing diagnostic biases and degrading care for patients with rare diseases.

To mitigate these harms, the machine learning community must avoid optimizing solely for visual fidelity, especially in the domain of medical image analysis. Furthermore, models should be evaluated in a multi-faceted manner (visual realism, privacy risks, and downstream utility). Additionally, our research shows a major blind spot in current privacy laws like HIPAA. We can no longer assume that synthetic data is automatically anonymous. To safely use AI models trained on private health data, we need strict access limits, regular privacy checks, and new rules that treat highly realistic AI images with the exact same care as real patient records.

**Acknowledgments**

Raman Dutt and Yongchen Yao are supported by the United Kingdom Research and Innovation (grant EP/S02431X/1), UKRI Centre for Doctoral Training in Biomedical AI at the University of Edinburgh. P. Sanchez thanks additional financial support from the School of Engineering, the University of Edinburgh. S.A. Tsaftaris acknowledges the UK's Engineering and Physical Sciences Research Council (EPSRC) (grant EP/X017680/1) and the UKRI AI programme and EPSRC, for CHAI - EPSRC AI Hub for Causality in Healthcare AI with Real Data (grant EP/Y028856/1). This work was also supported by the Edinburgh International Data Facility (EIDF) and the Data-Driven Innovation Programme at the University of Edinburgh. Access to EIDF was facilitated through the University of Edinburgh's Generative AI Laboratory GAIL Fellow scheme.

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

# A   Appendix

## A.1   Unreliability in Fidelity Estimates with Outdated Backbones

We demonstrate that existing protocols for evaluating generative fidelity in radiographic imaging are unreliable. Current approaches (Bluethgen et al., 2024; Dutt et al., 2024b; Lee et al., 2024) calculate image fidelity (FID Score) using an in-domain DenseNet-121 model trained on the MIMIC-CXR dataset. We argue this model lacks sufficient discriminative power, resulting in less meaningful fidelity assessments.

In our CheXGenBench benchmark, we address this limitation by leveraging features from RadDino (Pérez-García et al., 2025), a state-of-the-art model specifically designed for radiographs. As shown in Table 7, FID evaluation with DenseNet-121 shows minimal variance across models, often ranking several models at the same position. In contrast, the RadDino encoder significantly enhances evaluation quality by providing more meaningful feature representations that better differentiate between model performances.

Table 7: Comparison of FID and KID metrics across models using two distinct in-domain encoders. While the conventional DenseNet-121 encoder (Cohen et al., 2022) exhibits minimal variance (0.001 - 0.075) across models, indicating limited discriminative power, the RadDino encoder (Pérez-García et al., 2025) demonstrates substantially greater metric differentiation (54.22 - 194.72), providing more meaningful evaluation of model performance.

|  | Metric | SD V1-4 | SD V1-5 | SD V2 | SD V2-1 | RadEdit | Pixart Sigma | Sana | SD V3-5 | Lumina 2.0 | Flux.1-Dev | LLM-CXR |
|---|---|---|---|---|---|---|---|---|---|---|---|---|
| **FID** | DenseNet | 0.025 | 0.075 | 0.053 | 0.056 | 0.001 | 0.001 | 0.001 | 0.002 | 0.001 | 0.001 | 0.025 |
|  | RadDino | 125.180 | 118.930 | 194.720 | 186.530 | 69.690 | 60.150 | 54.220 | 91.300 | 101.190 | 122.400 | 71.240 |
| **KID** | DenseNet | 0.004 | 0.004 | 0.005 | 0.005 | 0.001 | 0.001 | 0.001 | 0.004 | 0.003 | 0.006 | 0.003 |
|  | RadDino | 0.172 | 0.147 | 0.376 | 0.413 | 0.033 | 0.023 | 0.016 | 0.103 | 0.110 | 0.144 | 0.061 |

## A.2   Ranking Text-to-Image Models

**Image Fidelity:** We present the rank for each T2I model across each individual fidelity and mode coverage metric in Tab. 8. We also present the combined rank averaged across all the metrics resulting in Sana (Xie et al., 2025), Pixart Sigma (Chen et al., 2024), and LLM-CXR (Lee et al., 2024) as the top-three performers across all models.

Table 8: Performance ranking of generative models for image fidelity across multiple evaluation metrics (lower rank indicates better performance). The top-3 performers are **(1)** Sana (Xie et al., 2025), **(2)** Pixart Sigma (Chen et al., 2024), and **(3)** LLM-CXR (Lee et al., 2024).

| Model | FID | | KID | | Alignment Score | Precision | Recall | Density | Coverage | Average Rank | Normalized Rank |
|---|---|---|---|---|---|---|---|---|---|---|---|
| | Inception | RadDino | Inception | RadDino | | | | | | | |
| **SD V1-4** | 9 | 9 | 9 | 9 | 4 | 8 | 5 | 7 | 8 | 7.55 | **8** |
| **SD V1-5** | 8 | 7 | 8 | 8 | 5 | 6 | 4 | 6 | 6 | 6.44 | **6** |
| **SD V2** | 10 | 11 | 10 | 10 | 7 | 9 | 6 | 9 | 10 | 9.11 | **10** |
| **SD V2-1** | 11 | 10 | 11 | 11 | 8 | 7 | 7 | 8 | 11 | 9.33 | **11** |
| **RadEdit** | 3 | 3 | 3 | 3 | 3 | 11 | 2 | 10 | 5 | 4.78 | **4** |
| **SD V3-5** | 5 | 5 | 5 | 5 | 10 | 2 | 11 | 4 | 7 | 6.00 | **5** |
| **Lumina 2.0** | 6 | 6 | 6 | 6 | 9 | 5 | 9 | 5 | 9 | 6.78 | **7** |
| **Flux.1-Dev** | 7 | 8 | 7 | 7 | 11 | 10 | 10 | 11 | 4 | 8.33 | **9** |
| **LLM-CXR** | 4 | 4 | 4 | 4 | 6 | 1 | 8 | 1 | 3 | 3.89 | **3** |
| **Pixart Sigma** | 2 | 2 | 2 | 2 | 1 | 4 | 3 | 3 | 2 | 2.33 | **2** |
| **Sana** | 1 | 1 | 1 | 1 | 2 | 3 | 1 | 2 | 1 | 1.44 | **1** |

Note: Lower rank numbers indicate better performance. Top three models highlighted based on normalized rank.

## A.3   Correlation Between Fidelity and Disease Distribution

In this section, we examine whether generative fidelity performance for individual pathologies, as reported in Tab. 2, correlates with the frequency of pathology occurrence in the training dataset. Conditions such as "No Finding (NF)", "Pleural Effusion (PE)", "Support Devices (SD)", and "Lung Opacity (LO)" represent

Table 9: Occurrence Frequency and Generative Fidelity for Different Pathologies with rankings. We calculate the "Fidelity Rank" across models from Tab. 2

| Pathology Code | Count (n) | Prevalence Rank | FID (RadDino) | Fidelity Rank |
|---|---|---|---|---|
| NF | 78,939 | 1 | 110.75 | 2 |
| SD | 71,537 | 2 | **109.77** | 1 |
| PE | 56,433 | 3 | 130.45 | 5 |
| LO | 53,513 | 4 | 133.77 | 7 |
| AT | 47,704 | 5 | 114.28 | 3 |
| CM | 46,602 | 6 | 114.89 | 4 |
| ED | 28,601 | 7 | 130.75 | 6 |
| PN | 16,832 | 8 | 146.69 | 8 |
| CD | 11,290 | 9 | 172.14 | 9 |
| PT | 10,971 | 10 | 172.15 | 10 |
| EC | 7,454 | 11 | 188.95 | 11 |
| LL | 6,491 | 12 | 194.99 | 12 |
| Frac. | 4,671 | 13 | 211.58 | 13 |
| PO | 2,024 | 14 | 227.43 | 14 |

**Note:** Lower FID (Fréchet Inception Distance) scores indicate better generative fidelity. The table shows a correlation between pathology prevalence and generative quality. Pathology codes: NF = No Finding, SD = Support Devices, PE = Pleural Effusion, LO = Lung Opacity, AT = Atelectasis, CM = Cardiomegaly, ED = Edema, PN = Pneumonia, CD = Consolidation, PT = Pneumothorax, EC = Enlarged Cardiomediastinum, LL = Lung Lesion, Frac. = Fracture, PO = Pleural Other.

the most frequently observed pathologies in the training data. Conversely, "Fracture" and "Pleural Other" exhibit significantly lower occurrence frequencies.

Tab. 9 presents the comparative rankings according to occurrence frequency and FID scores. Analysis reveals a remarkably strong positive correlation coefficient of **0.947** between these rankings, providing compelling evidence that generative fidelity demonstrates substantial dependence on pathology occurrence frequency in the training distribution.

### A.4 Correlation Between Fidelity and Downstream Performance

In this section, we study how well does the upstream generative fidelity of T2I models correlate with the downstream performance on different tasks. The results are presented in Tab. 10.

As detailed in Tab. 10, we conducted a rank-based analysis to investigate the relationship between upstream generative fidelity (measured via FID rank) and downstream clinical utility (measured via classification and segmentation ranks). The results demonstrate a strong positive Pearson correlation for both classification (**0.77**) and segmentation (**0.78**), indicating that, broadly, improvements in generative fidelity yield better downstream performance. However, analyzing the specific rank distributions reveals critical, task-dependent nuances. For image segmentation, the downstream rank closely mirrors the fidelity rank; top-performing generative models like Sana and Pixart Sigma consistently achieve the highest segmentation utility, whereas lower-fidelity models suffer proportional drops in rank. In contrast, the classification task exhibits a distinct plateau effect. Several models with lower fidelity ranks (such as SD V1-4 and SD V1-5) still achieve a top classification rank of 1, tying with the state-of-the-art architectures. This empirical finding quantitatively reinforces the Texture-Topology Gap hypothesis discussed in Section 4: while pixel-sensitive tasks like segmentation strictly require high-fidelity microscopic textures, macroscopic classification tasks can effectively leverage the overarching topological structures that are preserved even in lower-fidelity synthetic data.

Table 10: Model ranking comparison across FID, classification, and segmentation tasks.

| Model | FID Rank | Classification Rank | Segmentation Rank |
|---|---|---|---|
| SD V1-4 | 8 | 1 | 6 |
| SD V1-5 | 6 | 1 | 6 |
| SD V2 | 10 | 4 | 9 |
| SD V2-1 | 11 | 4 | 8 |
| RadEdit | 4 | 1 | 2 |
| Pixart Sigma | 2 | 1 | 1 |
| Sana | 1 | 1 | 1 |
| SD V3-5 | 5 | 2 | 2 |
| Lumina 2.0 | 7 | 2 | 2 |
| Flux.1-Dev | 9 | 2 | 2 |
| LLM-CXR | 3 | 1 | 1 |

### A.5 Training Settings and Hyperparameters

The hyperparameters (learning rates) for Text-to-Image model training are provided in Tab. 11. Existing foundation models (RadEdit, LLM-CXR) were not re-trained and used as is. For evaluating downstream utility, the learning rates are provided in Tab. 12. In all Low-Rank Adaptation (LoRA) experiments, the scaling factor $\alpha$ was set to equal the rank ($\mathbf{r}$), following the recommended practice. Furthermore, we adopted the rank-stabalized version of LoRA (rsLoRA) (Kalajdzievski, 2023) which has shown to improve convergence during training.

Table 11: Hyperparameters used for Text-to-Image (T2I) model training.

| Model | Fine-Tuning | LR |
|---|---|---|
| **SD V1-4** | FFT | 5e-6 |
| **SD V1-5** | FFT | 5e-6 |
| **SD V2** | FFT | 5e-6 |
| **SD V2-1** | FFT | 5e-6 |
| **RadEdit** | N/A | - |
| **Pixart Sigma** | FFT | 2e-5 |
| **Sana** | FFT | 1e-4 |
| **SD V3-5** | LoRA (r-32) | 1e-4 |
| **Lumina 2.0** | LoRA (r-32) | 1e-4 |
| **Flux.1-Dev** | LoRA (r-32) | 1e-4 |
| **LLM-CXR** | N/A | - |

Table 12: Hyperparameters for the downstream evaluation tasks.

| Model | ResNet-50 (Classification) | U-Net (Segmentation) | LLaVA-Rad (RRG) |
|---|---|---|---|
| **Fine-Tuning** | FFT | FFT | LoRA |
| **Learning Rate** | 1e-4 | 5e-5 | 1e-4 |

## A.6 Ablations of LoRA

**Ablations on Rank:** Our ablation study on the LoRA rank for large models (>1B parameters), presented in Tab. 13, reveals that increasing the rank modestly improves generation fidelity. However, our key finding is that a fully fine-tuned smaller model (Sana, 0.6B) still significantly outperforms much larger models adapted with PEFT (e.g., SD V3.5, 2.5B; Flux.1-Dev, 1.2B). This result is crucial for the medical image analysis community, as it highlights that thorough adaptation of an efficient model can be more effective and accessible than resource-intensive scaling of larger architectures. For each experiment, the scaling factor alpha ($\alpha$) was set equal to the rank following the recommended practice.

Table 13: Impact of LoRA Rank on FID Score.

| Model | FID by LoRA Rank | | |
|---|---|---|---|
| | **Rank 32** | **Rank 64** | **Rank 128** |
| **SD v3.5** | 91.30 | 84.14 | **74.58** |
| **Lumina 2.0** | 101.19 | 96.51 | **88.28** |
| **Flux.1-Dev** | 122.40 | 105.28 | **95.17** |
| **Average** | 104.96 | 95.31 | **85.68** |

Table 14: Impact of LoRA Position on FID Score (r=32, alpha=32)

| Metric/Layer (Fixed r=32) | Attn. | Attn. + MLP | All Layers |
|---|---|---|---|
| **FID** | 91.30 | 85.26 | 99.19 |

**Ablations on LoRA Position:** We conducted additional ablation studies on the placement of LoRA modules on SD v3.5. Specifically, we used a fixed rank (**r**) and alpha ($\alpha$) of 32 and explored two additional positions beyond the standard Attention Layers: **(1)** Attention + MLP layers and **(2)** Attention + MLP + Positional Embedding. The results are presented in Tab. 14.

The ablation study indicates that, for the fixed rank **r = 32**, none of the tested LoRA placement strategies yielded a significant improvement over the baseline placement in the Attention layers alone. While integrating LoRA into both the Attention and MLP layers showed a minor improvement, including the Positional Embedding layer resulted in a notable degradation of performance. These results suggest that applying LoRA to the Positional Embedding may disrupt vital spatial information. Future work should investigate whether substantial gains can be achieved by applying the most promising configurations (Attention + MLP) with a higher LoRA rank (e.g., $r = 128$)

### A.7 Exploring the Standalone Utility of Synthetic Data

In section 3.3, we explored the utility of augmenting real training set with synthetic samples. In this section, we focus on exploring the stand-alone utility of synthetic data, i.e. training models exclusively with synthetic data for image classification and report generation tasks.

#### A.7.1 Image Classification

Table 15: Performance Comparison (AUC ↑) of a ResNet50 classifier trained **only on synthetic data** vs. Original (Real) Data Baseline for all pathologies in the MIMIC dataset. Results matching or exceeding the Original Data baseline are **bolded** and within 0.01 AUC are underlined.

| Model | Atel-ectasis | Cardio-megaly | Consol-idation | Edema | EC | Fracture | LL | LO | NF | PE | PO | PN | PT | SD |
|---|---|---|---|---|---|---|---|---|---|---|---|---|---|---|
| **Original (Real)** | 0.75 | 0.76 | 0.72 | 0.85 | 0.61 | 0.58 | 0.63 | 0.70 | 0.84 | 0.84 | 0.74 | 0.67 | 0.71 | 0.83 |
| SD V1-4 | 0.70 | 0.70 | 0.67 | 0.81 | 0.56 | 0.57 | 0.63 | 0.67 | 0.80 | 0.77 | 0.65 | 0.60 | 0.65 | 0.80 |
| SD V1-5 | 0.72 | 0.72 | 0.69 | 0.81 | 0.60 | 0.53 | **0.66** | 0.67 | 0.82 | 0.79 | 0.68 | 0.62 | 0.70 | **0.83** |
| SD V2 | 0.66 | 0.69 | 0.66 | 0.78 | **0.61** | 0.53 | 0.55 | 0.63 | 0.75 | 0.76 | 0.50 | 0.61 | 0.64 | 0.78 |
| SD V2-1 | 0.63 | 0.67 | 0.65 | 0.71 | 0.55 | **0.59** | 0.62 | 0.62 | 0.75 | 0.74 | 0.57 | 0.56 | 0.61 | 0.75 |
| RadEdit | 0.73 | 0.73 | **0.72** | 0.84 | **0.61** | 0.56 | 0.60 | 0.69 | 0.81 | 0.82 | 0.72 | 0.66 | 0.66 | 0.77 |
| Pixart Sigma | 0.74 | 0.73 | 0.70 | 0.84 | **0.61** | **0.58** | 0.61 | 0.69 | 0.83 | 0.81 | 0.68 | 0.63 | 0.70 | 0.80 |
| Sana | 0.74 | **0.76** | **0.72** | **0.85** | **0.61** | **0.62** | **0.63** | **0.70** | 0.83 | **0.84** | 0.73 | 0.64 | **0.72** | **0.83** |
| SD V3-5 | 0.55 | 0.55 | 0.56 | 0.55 | 0.47 | 0.47 | 0.47 | 0.53 | 0.60 | 0.54 | 0.58 | 0.49 | 0.55 | 0.71 |
| Lumina 2.0 | 0.46 | 0.48 | 0.52 | 0.51 | 0.46 | 0.57 | 0.53 | 0.52 | 0.59 | 0.55 | 0.57 | 0.49 | 0.50 | 0.71 |
| Flux.1-Dev | 0.41 | 0.41 | 0.44 | 0.40 | 0.44 | 0.52 | 0.48 | 0.42 | 0.40 | 0.38 | 0.50 | 0.48 | 0.44 | 0.67 |
| LLM-CXR | 0.70 | 0.69 | 0.70 | 0.81 | **0.61** | 0.57 | 0.54 | 0.65 | 0.80 | 0.77 | 0.66 | 0.61 | 0.63 | 0.73 |

**Experimental Setup:** We measure the classification performance when a classifier (ResNet50) (He et al., 2016) is trained *exclusively* on synthetic data (20K samples) ($\mathcal{D}_{syn}$) (for 20 epochs). This provides us with an idea of the stand-alone clinical value of the synthetic data from a generative model. The performance metrics are calculated on a held-out real test set ($\mathcal{D}_{test}$) to ensure clinical relevance and generalizability of our findings. We quantify classification performance through standard accuracy, F1-Score, and AUROC metrics.

**Results:** We present the results in Tab. 15. Sana emerges as exceptionally effective, with its synthetic images enabling classifiers to match or exceed the original data baseline on an impressive 10 out of 13 pathologies. Interestingly, it surpasses the baseline on *Fracture*, an under-represented class in MIMIC-CXR. Other models, such as RadEdit, Pixart-Sigma, and LLM-CXR, show limited success by matching the baseline for at most two pathologies, failing to surpass it. Models like SD V1-4, SD V3-5, Lumina 2.0, and Flux.1-Dev generally produce synthetic data that leads to classifiers significantly underperforming the Original Data baseline across most or all pathologies. **Viability:** The results from Sana strongly suggest that high-quality synthetic data can, in some cases, be a viable standalone replacement for real data for training medical image classifiers. This is a powerful finding with implications for data privacy, scarcity, and augmentation.

#### A.7.2 Radiology Report Generation

Table 16: Radiology Report Generation (RRG) performance metrics for various generative models for *synthetic-only* experiments.

| Metric | Original | SD V1-4 | SD V1-5 | SD V2 | SD V2-1 | RadEdit | Pixart-Σ | Sana | SD V3-5 | Lumina 2.0 | Flux.1-Dev | LLM-CXR |
|---|---|---|---|---|---|---|---|---|---|---|---|---|
| **BLEU-1** (↑) | **38.16** | 25.85 | 26.02 | 26.62 | 26.76 | 30.55 | **31.25** | 31.11 | 23.49 | 17.96 | 18.19 | 29.78 |
| **BLEU-4** (↑) | **15.38** | 6.76 | 7.50 | 7.35 | 7.38 | **8.36** | 7.91 | 7.70 | 4.91 | 4.27 | 3.36 | 7.93 |
| **ROUGE-L** (↑) | **0.31** | 0.23 | **0.24** | **0.24** | **0.24** | **0.24** | 0.23 | **0.24** | 0.20 | 0.20 | 0.18 | 0.23 |
| **F1-RadGraph** (↑) | **0.29** | 0.21 | **0.24** | 0.23 | 0.22 | **0.24** | 0.22 | 0.23 | 0.17 | 0.18 | 0.14 | 0.21 |
| **Micro F1-5** (↑) | **0.57** | 0.56 | 0.54 | **0.57** | **0.57** | 0.55 | 0.50 | **0.57** | 0.31 | 0.41 | 0.32 | 0.55 |
| **Micro F1-14** (↑) | **0.57** | 0.53 | 0.51 | **0.55** | 0.53 | **0.55** | 0.52 | **0.55** | 0.35 | 0.41 | 0.36 | 0.53 |
| **GREENScore** (↑) | **0.36** | 0.28 | 0.27 | 0.27 | 0.27 | 0.30 | 0.30 | **0.31** | 0.28 | 0.28 | 0.27 | 0.30 |
| **RaTEScore** (↑) | **0.58** | 0.46 | 0.46 | 0.44 | 0.43 | 0.49 | 0.50 | **0.51** | 0.45 | 0.45 | 0.44 | 0.50 |

**Experimental Setup:** We adopt the pre-trained LLaVA-Rad model and continue to fine-tune it with 20,000 synthetic samples. The performance is reported on a real test set ($\mathcal{D}_{test}$).

| Model | Real + Syn. | Atelec. | C.Megaly | Consol. | Edema | En. C. | Frac. | LL | LO | NF | PE | PO | PN | PT | SD | Average |
|---|---|---|---|---|---|---|---|---|---|---|---|---|---|---|---|---|
| Sana | 20K + 5K | 0.670 | 0.690 | 0.700 | 0.790 | 0.620 | 0.590 | 0.610 | 0.680 | 0.830 | 0.810 | 0.710 | 0.610 | 0.730 | 0.830 | 0.705 |
| Sana | 20K + 10K | 0.700 | 0.720 | 0.720 | 0.820 | 0.630 | 0.620 | 0.630 | 0.700 | 0.830 | 0.830 | 0.730 | 0.640 | 0.740 | 0.840 | 0.725 |
| Sana | 20K + 20K | 0.760 | 0.780 | 0.740 | 0.870 | 0.660 | 0.630 | 0.700 | 0.730 | 0.850 | 0.860 | 0.770 | 0.690 | 0.780 | 0.860 | 0.760 |
| SD V1-4 | 20K + 5K | 0.740 | 0.750 | 0.730 | 0.840 | 0.640 | 0.590 | 0.730 | 0.720 | 0.830 | 0.840 | 0.720 | 0.680 | 0.740 | 0.800 | 0.739 |
| SD V1-4 | 20K + 10K | 0.740 | 0.760 | 0.750 | 0.860 | 0.650 | 0.590 | 0.730 | 0.730 | 0.840 | 0.840 | 0.720 | 0.700 | 0.760 | 0.850 | 0.751 |
| SD V1-4 | 20K + 20K | 0.760 | 0.780 | 0.740 | 0.870 | 0.650 | 0.600 | 0.730 | 0.730 | 0.850 | 0.850 | 0.740 | 0.700 | 0.770 | 0.870 | 0.760 |

Table 17: Downstream classification utility when scaling synthetic data volume from Sana and SD V1-4.

| Model | Real + 3K Syn. | Real + 7K Syn. |
|---|---|---|
| SD V1-4 | 0.593 | 0.575 |
| SD V1-5 | 0.586 | 0.566 |
| SD V2 | 0.537 | 0.512 |
| SD V2-1 | 0.541 | 0.537 |
| RadEdit | 0.667 | 0.671 |
| Pixart Σ | 0.671 | 0.679 |
| Sana | 0.667 | 0.679 |
| SD V3-5 | 0.612 | 0.601 |
| Lumina 2 | 0.581 | 0.556 |
| Flux.1 | 0.562 | 0.531 |
| LLM-CXR | 0.642 | 0.633 |

Table 18: Downstream segmentation utility when scaling synthetic data volume (3K vs 7K synthetic samples added to a 3K real baseline).

**Results:** The results are presented in Tab. 16. Firstly, we observe that *additional* fine-tuning with synthetic data, irrespective of the model, leads to a performance degradation as compared to the original baseline (trained with real data). In terms of the models, RadEdit and Sana emerge as leading performers. RadEdit excels in BLEU-4 (fluency) and is a top contender in F1-RadGraph and Micro F1-14, while Sana demonstrates strengths in ROUGE-L and Micro F1 scores for identifying specific findings. LLM-CXR also gives a strong performance, often giving second- or third-best scores. Pixart Sigma shows the best individual word usage (BLEU-1), and models like SD V2 also perform well in Micro F1 scores. Overall, no single model dominates in all metrics. **Explanation:** These results reflect that for a multi-modal task such as RRG, fine-tuning solely with synthetic data degrades performance for both low and high fidelity T2I models. We explain this through the Texture-Topology gap previosuly discussed in Section 4. LLaVA-Rad was originally trained with real radiographs. When continually fine-tuned with synthetic X-rays, the model "forgets" its original knowledge due to distinct differences between real and synthetic radiographs. This forgetting results in performance degradation when evaluated on a real dataset.

## A.8   Scaling Synthetic Data for Downstream Utility

To further investigate the relationship between synthetic data volume and downstream utility, we conducted additional ablation experiments scaling the number of synthetic samples used for augmentation. Specifically, we compared a high-fidelity model (Sana) against a low-fidelity model (SD V1-4) across both image classification and image segmentation tasks, varying the synthetic data volume added to the real training set (e.g., 10K, 20K, 30K, and 40K samples).

**Scaling in Image Classification:** For the classification task, which predominantly relies on macroscopic topology, we observed that scaling synthetic data volume provides performance gains for both Sana and SD V1-4 (Tab. 17). However, the utility scales differently between the two architectures. While the high-fidelity images from Sana offer consistent improvements across all scaling thresholds, the low-fidelity images from SD V1-4 primarily serve as a robust regularizer, with performance gains plateauing at higher data volumes.

**Scaling in Image Segmentation:** In contrast, the image segmentation task, which is highly sensitive to microscopic texture and sharp local gradients, demonstrates a stark difference (Tab. 18). Scaling the synthetic

data from Sana yields consistent, incremental improvements in segmentation accuracy, confirming that high-quality synthetic textures reliably enhance pixel-level representations at scale. Conversely, scaling the data from SD V1-4 results in stagnant or even degraded performance at higher volumes. The "waxy" smoothing characteristic of these low-fidelity generations exacerbates the domain shift when scaled, demonstrating that synthetic data augmentation only provides consistent scaling for texture-sensitive tasks when the generative fidelity is sufficiently high.

These scaling experiments reinforce our findings regarding the Texture-Topology Gap (Section 4), underscoring that while low-fidelity data can provide structural regularization for global tasks, true pixel-level scaling requires models that accurately capture high-frequency microscopic details.

## A.9 Benefits of Adopting LLaVA-Rad Annotations

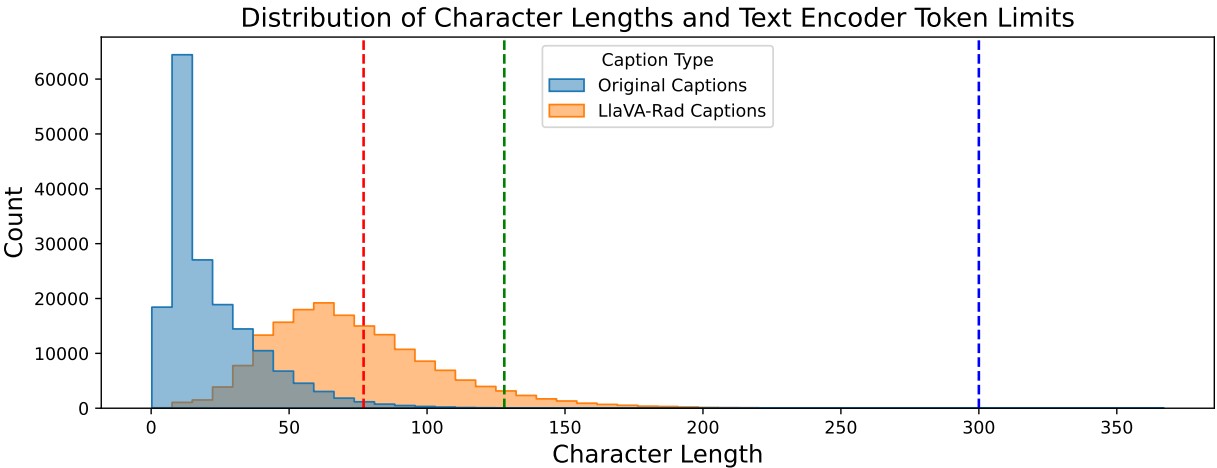

Figure 3: Figure depicting distribution of character lengths for original MIMIC captions and LlaVA-Rad annotations. We also illustrate the text-encoder token length limits for all SD variants and Flux (77 tokens), RadEdit (128 tokens), and Pixart Sigma (300 tokens).
Note: We treat 1 token ≈ 4 characters[2]

Table 19: Comparison of generation fidelity (left) and privacy preservation (right) metrics across different stable diffusion models. LlaVA-Rad annotations consistently outperform original MIMIC impressions, yielding improved image quality (FID/KID (↓), higher alignment (↑)) and enhanced privacy protection (Re-ID (↓), latent/pixel distances (↑)).

| Model | Prompt Type | FID (RadDino) | KID (RadDino) | Alignment Score |
|---|---|---|---|---|
| **SD-V1-4** | Original MIMIC | 147.298 | 0.198 | 0.272 |
| **SD-V1-4** | LlaVA-Rad | 125.186 | 0.172 | 0.357 |
| **SD-V1-5** | Original MIMIC | 144.661 | 0.201 | 0.257 |
| **SD-V1-5** | LlaVA-Rad | 118.932 | 0.147 | 0.326 |
| **SD-V2** | Original MIMIC | 214.202 | 0.496 | 0.145 |
| **SD-V2** | LlaVA-Rad | 194.724 | 0.376 | 0.311 |

(a) Generation fidelity metrics.

| Model | Prompt Type | Max. Re-ID Distance↓ | Min Latent Distance↑ | Min. Pixel Distance↑ |
|---|---|---|---|---|
| **SD-V1-4** | Original MIMIC | $0.725 \pm 0.71$ | $0.482 \pm 0.66$ | $132.24 \pm 4.6$ |
| **SD-V1-4** | LlaVA-Rad | $0.539 \pm 0.31$ | $0.592 \pm 0.05$ | $143.44 \pm 4.6$ |
| **SD-V1-5** | Original MIMIC | $0.721 \pm 0.47$ | $0.476 \pm 0.41$ | $131.44 \pm 4.2$ |
| **SD-V1-5** | LlaVA-Rad | $0.572 \pm 0.29$ | $0.583 \pm 0.04$ | $143.634 \pm 4.2$ |
| **SD-V2** | Original MIMIC | $0.687 \pm 0.23$ | $0.483 \pm 0.34$ | $132.64 \pm 4.3$ |
| **SD-V2** | LlaVA-Rad | $0.533 \pm 0.32$ | $0.588 \pm 0.05$ | $143.936 \pm 4.3$ |

(b) Privacy and memorisation risk metrics.

In this section, we demonstrate that LLaVA-Rad Annotations lead to substantial improvements in both fidelity performance and reduction of re-identification risks (Table 19). We attribute these improvements to two key factors: the enhanced descriptiveness of the annotations and the removal of certain tokens from the original MIMIC annotations known to increase privacy risks (Dutt, 2025). Figure 3 displays the distribution

---

[2]https://help.openai.com/en/articles/4936856-what-are-tokens-and-how-to-count-them

of average character lengths across both annotation types. MIMIC annotations cluster around significantly smaller values, while LLaVA-Rad Annotations exhibit a wider distribution, indicating greater descriptive detail. Table 19a reveals that LLaVA-Rad Annotations significantly enhance all three fidelity metrics (FID, KID, and Image-Text Alignment) compared to the original MIMIC annotations. Additionally, in Tab. 19b, we observe a substantial improvement in privacy risk mitigation, further validating the superiority of the LLaVA-Rad annotation approach.

