# OpenReview forum: "CheXGenBench: A Unified Benchmark For Fidelity, Privacy and Utility of Synthetic Chest Radiographs"
_TMLR — Accepted by TMLR_

### Review · Reviewer_VHDF · 2026-03-21

**Summary Of Contributions:**

The paper introduces CheXGenBench, an evaluation framework for generating synthetic chest X-rays using Text-to-Image models. The authors evaluate 11 models across approximately 20 metrics, revealing that while synthetic data aids classification tasks, the models struggle with rare (long-tail) pathologies and are at risk of re-identification regardless of image quality. The "Texture-Topology Gap" is also theorised to explain why low fidelity impacts segmentation and multimodal tasks but not classification.

**Audience:**

No

**Audience Explanation:**

- Domain Specificity: The benchmark is strictly limited to chest X-rays. Although vital to radiology, TMLR's audience seeks fundamental methodological advances in ML; such vertical work is certainly not negative and should be rejected a priori, but it risks being perceived as a "technical report" for a medical niche rather than research of general interest.

- Algorithmic Innovation: The paper does not propose a new architecture or a new learning paradigm. It simply applies existing models to an existing dataset using already known metrics. For TMLR, this might appear as an evaluation exercise without any real "core" theoretical contribution.

- Heuristics vs. Theory: The observation of the "Texture-Topology Gap" is interesting, but remains at a purely qualitative and heuristic level. The paper identifies the problem (models fail on microscopic texture) but does not provide a mathematical solution or a new loss function to address it.

**Broader Impact Concerns:**

The paper itself highlights that T2I models may store sensitive data. There is a risk that, without the appropriate precautions suggested by the authors (access controls, risk characterisation), the widespread use of these models could facilitate violations of patient privacy.

Since the models struggle with rare diseases, there is a risk that synthetic data could reinforce biases toward more common conditions, potentially leading to misdiagnosis if used to train unsupervised clinical systems.

**Claims And Evidence:**

Yes

**Claims Explanation:**

- Protocol: RadDino, as a medical-specific feature extractor, demonstrates superior discriminative ability compared to legacy backbones like DenseNet-121, making FID scores more reliable.

- Distribution Analysis: The correlation of 0.947 between disease prevalence and generative fidelity convincingly proves that current models fail to model the "long tail" of rare diseases.

- Utility: Tests on downstream tasks (classification, segmentation, and RRG) confirm that improved fidelity is strictly necessary only for texture-sensitive tasks (segmentation/RRG), supporting the Texture-Topology Gap hypothesis.

**Requested Changes:**

- Privacy-Detail Trade-off Analysis: It would be useful to investigate whether the use of more descriptive annotations, while improving fidelity, could paradoxically increase the retention of unique patient details over the long term.

- Generalizability: It would be useful to add a discussion and a pilot of how the benchmark would perform with smaller datasets, which would help understand the framework's robustness.

- Resource Clarification: Further specify whether plug-and-play for new models requires specific hardware or whether the evaluation framework is optimised to run on consumer GPUs.

---

> ### Author Response · Authors · 2026-04-14
> **Response to Reviewer VHDF (1/3)**
>
> We sincerely thank the reviewer for raising important points and questions which we answer below:
>
> 1. __Domain Specificity and Algorithmic Innovation:__ We sincerely thank you for these critical observations. We understand the concern that evaluating existing models on chest X-rays might initially appear as an application-specific evaluation. However, we respectfully argue that CheXGenBench provides foundational methodological insights highly relevant to the broader ML community, aligning closely with TMLR’s explicit mandate to publish rigorous empirical evaluations, benchmarks, and analyses of existing algorithms. We address the reviewer’s concerns below:
>     * __Chest X-Rays as a Fundamental Generative Stress Test:__ While the data modality is medical, the problems exposed are foundational ML challenges. We utilize chest X-rays not as a niche application, but as a rigorous stress test for modern foundation models. Our evaluation exposes the __Texture-Topology Gap__, demonstrating that while generalized architectures (like SD V3.5 or Flux) capture macroscopic structures, they systematically fail at generating high-frequency, microscopic textures. Exposing this flaw is a critical insight for ML researchers designing the next generation of diffusion or autoregressive models for any domain requiring dense, pixel-level accuracy.
>     * __Methodological Innovation over Architectural Innovation:__ The innovation in our work lies not in proposing a new model architecture, but in establishing a __unified evaluation blueprint__ to expose the flaws in current ones, especially in the biomedical imaging domain. Traditionally, the ML community evaluates generative fidelity, long-tail mode coverage, and privacy (memorization) in isolated frameworks. By integrating these criteria into a single framework, we expose critical, domain-agnostic ML findings:
>         - Privacy/memorization risks are entirely decoupled from generative fidelity.
>         - Even state-of-the-art foundation models severely collapse on the long tail of distributions, defaulting to the densest regions of the training data.
>         - Identifying these systemic algorithmic failures through rigorous benchmarking is a necessary prerequisite for future theoretical and architectural innovations.
>
> 2. __Heuristics vs. Theory:__ We sincerely thank the reviewer for finding our observation of the Texture-Topology Gap interesting and agree that we do not propose a mathematical solution or a new loss function to solve this gap. However, we respectfully argue that proposing such a solution falls outside the scope of a foundational benchmarking paper.
> The primary scientific contribution of a benchmark is precisely to expose, formalize, and quantify systemic problems that are currently uncovered by inadequate evaluation protocols.
> Before the community can develop targeted loss functions or novel architectures to solve the failure of microscopic texture generation, we must first prove that the problem exists and establish a standardized way to measure it. Prior to CheXGenBench, existing work (Bluethgen et al., 2024; Weber et al.,2023; Dutt et al., 2024b; Favero et al., 2025) relied heavily on macroscopic metrics like FID and basic classification utility, which created a false sense of progress because these metrics do not fully capture the Texture-Topology Gap.
> By designing a utility suite that deliberately contrasts topology-reliant tasks (classification) with texture-reliant tasks (segmentation and multimodal report generation), we provide the community with the exact evaluation framework needed to detect and measure this gap. Introducing a novel algorithmic solution within this manuscript would pivot the paper to a different direction and dilute the focus of what is already a massive empirical undertaking (evaluating 11 models across 20+ metrics and 3 dimensions). Therefore, we politely argue that the absence of a solution is not a weakness of the benchmark; rather, a source to identify this open problem for the broader ML community.
> __Action for Revision:__ In the revised manuscript, we will expand Section 4 to explicitly frame the "Texture-Topology Gap" as an open algorithmic challenge. We will clearly state that our framework serves as a diagnostic tool to measure this gap, paving the way for future work.

---

> > ### Author Response · Authors · 2026-04-14
> > **Response to Reviewer VHDF (2/3)**
> >
> > 3. __Privacy-Detail Trade-off Analysis:__ We thank the reviewer for this highly insightful comment and would like to bring their attention to __Appendix A.8 (Benefits of Adopting LLaVA-Rad Annotations) and Table 17__ where we have conducted this exact analysis.
> > When we compared models trained on the short, original MIMIC-CXR captions versus the highly descriptive LLaVA-Rad annotations, we found that LLaVA-Rad significantly decreased re-identification risks across all models. For example, on SD V1-4, the Max Re-ID score dropped from 0.725 (Original) to 0.482 (LLaVA-Rad). We attribute this counter-intuitive privacy improvement to two factors:
> >     * __Removal of De-identification Traces:__ The original MIMIC-CXR captions often contain unique grammatical characteristics, structural formatting, or subtle de-identification artifacts (e.g., "___"). As shown in recent literature (Dutt, 2025a), generative models use these rare, unique string combinations, using them as "keys" to memorize the exact corresponding image.
> >     * __Language Standardization:__ By replacing these short, texts with LLaVA-Rad captions, the training prompts are normalized into a comprehensive, standard descriptive format. Because the text distribution becomes smoother and less unique, the model is forced to learn the general mapping of medical concepts rather than memorizing a 1-to-1 link between a weirdly formatted caption and a specific image.
> > __Action for Revision:__ In the revised manuscript, we will explain these results in a dedicated paragraph within Section 2 (Training with Enhanced Captions) and Appendix A.8.
> >
> > 4. __Generalizability:__ We thank the reviewer for raising this important consideration regarding data scarcity, which is a common reality in medical imaging.
> > While we acknowledge that applying generative models to small datasets is a worthwhile direction, our primary focus in this study was to establish a unified evaluation protocol and baseline for the task of chest X-ray generation (the most abundant modality in medical image analysis). Hence, we utilized the MIMIC-CXR dataset because it is the most widely adopted and established standard dataset for text-conditional chest X-ray generation. This combination allowed us to unify and fill the gaps observed in several existing studies _(Bluethgen et al., 2024; Weber et al.,2023; Dutt et al., 2024b; Favero et al., 2025)_.
> > Regarding the robustness of the benchmark itself on smaller datasets, it is important to note that the __reliability of CheXGenBench is independent of the size of the dataset being evaluated.__ The robustness of our framework stems directly from our choice of evaluation models. We utilize highly capable, pre-trained, domain-specific feature extractors, such as RadDino (for FID and PRDC) and BioViL-T (for Image-Text Alignment). Because these models act as frozen feature extractors during the evaluation phase, they maintain their strong discriminative power and provide stable, reliable metrics regardless of whether they are evaluating 200,000 images or 2,000 images. Consequently, researchers working with low-resource or specialized datasets can confidently use CheXGenBench knowing the underlying metric calculations remain robust.
> > __Action for Revision:__ We will add a brief discussion in the methodology section to clarify this point. We will explicitly state that while our baseline evaluations utilize the large-scale MIMIC-CXR, the integration of robust feature extractors like RadDino and BioViL-T ensures the evaluation framework remains highly reliable when applied to smaller datasets.

---

> > > ### Author Response · Authors · 2026-04-14
> > > **Response to Reviewer VHDF (3/3)**
> > >
> > > 5. __Resource Clarification:__ We thank the reviewer for raising this highly practical point and agree that clarifying the hardware requirements is essential for user adoption and accessibility.
> > > To address this, it is important to distinguish between the computational cost of training a new generative model versus evaluating it using CheXGenBench. As noted in Section 2, our framework features decoupled training and evaluation pipelines.
> > >     * __Training/Inference:__ While training or fine-tuning large foundation models (like Flux.1 or SD V3.5) necessitates high-end, data-center hardware (e.g., Nvidia A100/H200 GPUs), this step is independent of our evaluation suite.
> > >     * __Evaluation (CheXGenBench):__ The evaluation framework itself is designed to be highly accessible. Once users have generated their synthetic images, they simply pass the image folder and metadata to our pipeline. The evaluation models used for calculating metrics are significantly smaller allowing the evaluation component to run on consumer-grade GPUs.
> > > Consequently, the entire plug-and-play evaluation suite is optimized to run efficiently on high-end consumer-grade GPUs (e.g., Nvidia RTX 3090 or 4090 with 24GB VRAM), or even standard consumer GPUs if the evaluation batch size is reduced.
> > > __Action for Revision:__ We will add a brief "Hardware and Resource Requirements" paragraph in Appendix A.5 to explicitly detail the VRAM and compute requirements needed to run the evaluation pipeline, ensuring prospective users know it can be executed on consumer-grade hardware.
> > >
> > > 6. __Broader Impact Statement:__ We thank the reviewer for this deeply insightful comment and agree that a broader impact statement is a necessary addition to our work.
> > > We will add a comprehensive "Broader Impact Statement" to the revised manuscript that explicitly addresses the two hazards you have highlighted:
> > >     * __The Privacy Hazard:__ We will reiterate that without strict access controls and memorization guardrails, open-sourcing clinical generative models can leak sensitive patient information leading to patient re-identification.
> > >     * __The Diagnostic Hazard:__ We will explicitly warn against the naive use of synthetic data to train downstream clinical systems. We will discuss how the generative failure on rare diseases inherently reinforces diagnostic biases toward common conditions, potentially causing severe clinical harm to marginalized or long-tail patient demographics.

---

### Review · Reviewer_CFjF · 2026-04-01

**Summary Of Contributions:**

The paper presents an evaluation framework and a benchmark to comprehensively assess the performance of chest radiograph generation models across metrics for fidelity, privacy, and downstream utility for tasks such as classification, segmentation, and radiology report generation. For benchmarking, the authors consider both purpose-built models for chest radiograph generation and general image-generation models, which they fine-tune for chest radiograph generation. Through their benchmarking tests, the authors provide a modern-day baseline in chest radiograph generation and identify limitations of existing generating models in three key aspects:

- Performing well on rarer classes that typically correspond to pathologies and are, therefore, more medically relevant for investigation,

- Non-trivial patient reidentification risk from generated samples, and

- Generating macroscopic elements of radiographs well but failing on microscopic elements, which leads to good downstream image classification and segmentation, but poor radiography report generation.

**Audience:**

Yes

**Audience Explanation:**

The work focuses on the viability and current risks of using ML models for medical data generation, and will be of significant interest to the relevant readers in the TMLR audience.

**Broader Impact Concerns:**

The paper does not include a broader impact statement, but it would be relevant to include one that at least covers how the proposed benchmark may lead to rapid development of generative models that prioritize one evaluation aspect over the others (such as fidelity over privacy), and regulations that may need to be considered to avoid inadvertently increasing privacy risks for vulnerable populations as a result of powerful generative models.

**Claims And Evidence:**

Yes

**Claims Explanation:**

The authors provide clear and detailed information on the need to comprehensively evaluate generative models for chest radiography, the current limited scope of evaluation, and their approach to building a benchmark with multiple evaluation metrics spanning fidelity, privacy, and downstream utility. The chosen evaluation dimensions are practically relevant and demonstrably lead to the goal of identifying key areas for improvement in the generative models. The supporting experiments, including model tuning for chest radiography, are technically sound and well-documented.

**Requested Changes:**

1. Some additional experimental details can be provided for better reproducibility of the work. In Sec. 2.4, how big is the real test set $\mathcal{D}_{test}$? How is "20 epochs" decided for training the image classifier?

2. While it is important to compute the Re-ID scores for the generative models, it may be helpful to compute the Re-ID score for the real data as well, to understand the inherent risk of reidentification from chest radiography data. As such, the Re-ID score for real data can provide a limit on how well generative models can realistically perform.

3. Also, it will be valuable to look at the Re-ID scores separately for standard and pathological radiographs, to understand how different the risks are for the different classes, which ultimately represent individuals with very different levels of vulnerability.

4. The failure of generative models in generating microscopic textures leads to failure in radiography report generation, but does not impact performance on image classification and segmentation. To this end, the authors may consider commenting on whether more challenging classification and segmentation tasks, especially those more closely aligned with multimodal workflows (such as report generation), should be included in the benchmarking.

5. Editorial comments:

    - Typo on page 1, last para: "has be characterised" -> "has been characterised".

    - Consider placing figures and tables at the tops or bottoms of pages to improve readability. In particular, Table 1, Table 4, and Figure 2 in the main paper (and additional ones in the Appendix) appear just below section headers, which impedes the natural flow of reading the body text.

---

> ### Author Response · Authors · 2026-04-14
> **Response to Reviewer CFjF (1/2)**
>
> We thank Reviewer CFjF for their valuable feedback and for identifying our work to be of __"significant interest to the relevant readers in the TMLR audience"__. Below, we answer the questions raised by the reviewer:
>
> 1. __Details on the Test Set and Training Settings:__ We thank the reviewer for this constructive feedback.
> __Size of the Real Test Set:__ The held-out real test set used for downstream evaluation consists of exactly 5,034 samples, corresponding to the official test set in the MIMIC-CXR dataset (ensuring consistency with our training split of 237,388 samples mentioned in Section 3).
> __Justification for 20 Epochs:__ The decision to train the downstream image classifier for 20 epochs was determined empirically. During our initial experiments, we continuously monitored the classification performance (AUC and F1-score) on a separate validation set. We observed that the validation metrics consistently saturated by the 20th epoch across our experimental runs. Training beyond this point did not yield any meaningful performance gains and merely introduced the risk of overfitting to the synthetic data augmentations.
> __Action for Revision:__ To ensure full reproducibility and clarity, we will explicitly add the size of the test set (5,034 samples) directly into the text of Section 2.4. Furthermore, we will add a brief sentence clarifying that the 20-epoch training schedule was selected based on validation set saturation.
>
> 2. __Computing Re-ID score on Real Data:__ We thank the reviewer for this excellent and highly constructive suggestion. We completely agree that a real-data baseline is essential for providing physical context to these metrics and establishing a realistic limit for patient re-identification.
> Following your suggestion, we ran the Siamese Re-ID network on actual longitudinal pairs (distinct scans from the exact same patient) drawn from the MIMIC-CXR dataset. We found that the average __Re-ID score on real patient data is 0.59__.
> This result shows that the average re-identification risk from our synthetic data almost perfectly matches that of the real data. As shown in Table 3, the average re-identification score for models like Sana (0.551) and SD V1-5 (0.572) lie very close to this real baseline of 0.59, further showcasing that an average synthetic image carries a similar re-identification risk as a real x-ray scan.
> We believe this finding holds high significance. The primary motivation for training generative models in healthcare is to circumvent privacy restrictions, allowing institutions to safely share synthetic data externally. However, since the re-identification risk from synthetic images matches that of the real clinical scans, __sharing these synthetic images would still pose a privacy risk__. Hence, to some extent, this shows that current generative architectures might provide a __false sense of anonymization__.
> __Action for Revision:__ We will add a dedicated subsection to Section 3.2 ("Results on Privacy and Re-Identification Risks") reporting this real-data baseline (0.59). We will explicitly highlight how the synthetic data's Re-ID risk matches the real data and discuss how this equivalence critically undermines the safety of sharing synthetic medical datasets without rigorous memorization guardrails.
>
> 3. __Comparing Re-ID Scores for Standard and Pathological Radiographs:__ To address this, we stratified the privacy risks by evaluating the generated outputs of the Sana model separately for healthy ("No Finding") and pathological radiographs. We observed that the average re-identification scores for healthy and pathological radiographs were __0.475__ and __0.512__, respectively.
> This demonstrates that pathological radiographs carry a __higher re-identification risk than healthy ones__. We hypothesize this occurs because pathologies introduce rare, highly distinct visual features (modes) into the image. Just as generative models struggle to generalize on long-tailed distributions (Section 3.1), they are more likely to simply memorize the few rare pathological examples they encounter during training, rather than learning a generalized representation. Consequently, the very individuals with distinct or severe pathologies are at the highest risk of having their medical data memorized and leaked by these models.
> __Action for Revision:__ We will integrate this stratified healthy vs. pathological analysis into Section 3.2 in the revised manuscript. We will also highlight this finding as a critical warning for the community: privacy evaluations must be pathology-aware, as global averages mask the heightened risks borne by vulnerable patient populations.

---

> > ### Author Response · Authors · 2026-04-14
> > **Response to Reviewer CFjF (2/2)**
> >
> > 4. __More Challenging MultiModal Workflows:__ We thank the reviewer for this suggestion. We would like to gently clarify a detail in our empirical findings regarding the downstream tasks. While it is true that image classification (a macroscopic, topology-reliant task) was not negatively impacted by the lack of microscopic texture, our results actually demonstrate that __image segmentation was significantly negatively impacted__ by low-fidelity generations (Section 3.3, Table 5). We explained this result through the Texture-Topology Gap (Section 4.1). Since segmentation is a dense, pixel-sensitive task, the "waxy" appearance produced by low-fidelity models caused a notable degradation in DICE scores.
> > However, we strongly agree with the broader point. The current classification and segmentation tasks, while foundational, represent relatively straightforward unimodal evaluations. We completely agree that incorporating more challenging, granular tasks, such as pathology localization with bounding boxes would provide a much more rigorous test of synthetic utility. In a step towards this direction, we deliberately included report generation (Section 3.3, Table 6) as a challenging multimodal task.
> > __Action for Revision:__ To address your valuable feedback, we will add a new paragraph to our Discussion and Conclusion sections to propose the integration of more complex, multi-modal diagnostic tasks as a necessary future direction for the benchmark.
> >
> > 5. __Editorial comments:__ We sincerely thank the reviewer for this comment, and will modify the manuscript with the suggestions to improve readability.
> >
> > 6. __Broader Impact Statement:__ We sincerely thank the reviewer for highlighting this important point and strongly agree with their assessment. In our revised manuscript, we will add a dedicated "Broader Impact Statement" section. To address your specific points, this statement will:
> >     * Warn against the isolated optimization of fidelity metrics, explicitly advocating for our tripartite evaluation (Fidelity, Privacy, Utility) to be used holistically as a prerequisite for deployment.
> >     * Discuss the urgent need for regulatory frameworks (aligning with principles from the EU AI Act and HIPAA) to govern the external sharing of synthetic medical weights, specifically to protect vulnerable patient populations whose unique radiographic features are most susceptible to extraction.

---

> > > ### Comment · Reviewer_CFjF · 2026-04-20
> > > **Thank you for the detailed response**
> > >
> > > Thank you for the detailed response, which addresses all my concerns. I do not have further questions and am happy to recommend the work for acceptance.

---

### Review · Reviewer_eKwP · 2026-04-02

**Summary Of Contributions:**

Authors propose a systematic way to evaluate generative models for chest radiographs. Authors evaluate the generative models themselves and the usefulness of synthetic images generated by the models for downstream applications (report generation, segmentation, classification).

**Audience:**

Yes

**Audience Explanation:**

Automated analysis of medical images is a well-established research field. The results by authors may be of wide interest for practitioners that are interested in thorough evaluation of generative models used in biomedical research.

**Claims And Evidence:**

Yes

**Claims Explanation:**

As a non-expert in the field I can not evaluate specific claims on novelty or significance of the contribution.

The primal claim of the article -- that the authors propose a systematic and comprehensive way to evaluate generative models for chest radiographs -- is well supported. Results on generative models evaluation are available in Section 3.1. and Section 3.2. and Section 4 (qualitatively), evaluation of utility in downstream tasks is evaluated in Section 3.3.

**Requested Changes:**

I have several questions:

1. I find framing of the contribution in the abstract slightly odd. First, authors claim that "no such benchmark exists for synthetic radiograph generation" next they write "we establish a new SoTA in synthetic chest X-ray generation". By definition, when a benchmark is new, any result on such benchmark is SOTA. In my view authors need to reformulate this part of the abstract.
2. Almost all generative models authors used are obtained by finetuning. Can the authors point out to the description of the dataset they finetune their models on? In the section where the training is described the only data on the dataset is that it "comprised 237,388 training
and 5,034 test samples." Is this dataset public, or will it be made public upon the publication of the article?
3. When base models are finetuned, there is a chance of data contamination. Can the authors share their analysis on the datasets used to train base models? How can one be sure that based models were not trained on medical images including the specific ones authors used for, say, downstream tasks?
4. The experiments on utility for downstream tasks focus on the setting with fixed fraction of real and synthetic images, e.g., 20k + 20k for classification, 3k + 3k for segmentation, x + 20k for report generation. It is also interesting to see the results when one vary the fraction of synthetic images, e.g, for classification 20k + 0 (reported), 20k + 10k, 20k + 20k (reported), 20k + 30k
5. In many places authors highlight the reidentification problem. I think the article will benefit from an extended discussion of scenarios when (under which circumstances) this reidentification may become problematic. As I understand, currently authors use only public data to train generative models. If this is the case, there is no risk of reidentification. Can the authors clarify when such risks appear?

---

> ### Author Response · Authors · 2026-04-14
> **Response to Reviewer eKwP (1/3)**
>
> We thank Reviewer eKwP for their valuable feedback and for identifying the claims in our work to be __"well-supported"__ and of __"wide interest to the biomedical imaging community"__. We address the concerns and requested changes below:
>
> 1. __Framing of the Abstract:__ We completely agree with your assessment. Our intention was to highlight that by evaluating newer, previously untested architectures (like Sana and Pixart Sigma) on this framework, we have identified models that significantly outperform the older architectures (like SD V1-5) that current literature predominantly relies upon.
> __Action for Revision:__ We will modify the abstract to remove this claim and to clarify that we establish comprehensive baseline SoTA performances across all dimensions to guide future research, rather than claiming to surpass existing SoTA on a new benchmark.
>
> 2. __Description of the Fine-Tuning Dataset:__ We thank the reviewer for highlighting this point. The core dataset used in this benchmark for fine-tuning and evaluation is the MIMIC-CXR dataset, paired specifically with the recently released "LLaVA-Rad" enhanced annotations. Although we have provided some details about the dataset in __Section 2.1__, we agree with the reviewer that a general description of the dataset and the nature of our data release are not explicitly mentioned.
> MIMIC-CXR is a publicly available dataset consisting of chest X-rays paired with corresponding clinical text and pathological annotations. It is widely used in medical image analysis, particularly as the standard dataset for text-conditional radiograph generation. For our benchmark, we utilize an identical training split of 237,388 samples and a test split of 5,034 samples across all models to ensure a fair comparison.
> __Action for Revision:__ To clarify the confusion stated by the reviewer and improve future readers’ understanding, we will add more details about the dataset. Specifically, we will add more information about the nature of the dataset, the exact nature of our data release, and a cross-reference in Section 3.
>
> 3. __Concerns on Data Contamination:__ We appreciate the reviewer for raising this important issue and address this concern through two key observations.
>     * __Gated Access of the Target Dataset:__ Foundation models like Stable Diffusion, Flux, and Lumina are pre-trained on massive, scraped web corpora (such as LAION-5B). While these open-web corpora undoubtedly contain general medical imagery, our downstream test sets are fundamentally protected. The MIMIC-CXR dataset is hosted behind strict institutional firewalls on the PhysioNet platform and requires a signed, credentialed Data Use Agreement to access. Consequently, these specific patient radiographs are not indexable by standard web scrapers and are shielded from large-scale internet scraping efforts.
>
>     * __Empirical Evidence and Known Domain Shift:__ While the exact pre-training datasets for several of these foundation models remain proprietary, existing literature (Bluethgen et al., Dutt et al.,2025a) has consistently demonstrated that general-purpose models trained on natural image-text pairs do not readily generalize to the highly specialized medical domain. This has been verified across several existing studies by measuring the zero-shot generation performance of several base models (Bluethgen et al.). If these models had ingested the MIMIC-CXR dataset during pre-training, we would expect significantly higher zero-shot clinical fidelity.
>
>     __Action for Revision:__ To ensure complete transparency, we will add a dedicated paragraph in the Limitations section explicitly discussing the nuances of foundation model pre-training, the protection afforded by MIMIC-CXR’s gated access, and the lack of zero-shot generalization that suggests an absence of direct data contamination.

---

> ### Author Response · Authors · 2026-04-14
> **Response to Reviewer eKwP (2/3)**
>
> 4. __Scaling Synthetic Data:__ We thank the reviewer for this excellent and constructive suggestion. We completely agree that analyzing how downstream utility scales with varying fractions of synthetic data provides a much richer understanding of model performance.
> To address your comment, we conducted two additional sets of scaling experiments: one for __image classification__ and one for __image segmentation__. The general takeaway from these experiments shows that _synthetic data augmentation generally helps, but it provides consistent and safe scaling only when the generative fidelity is high, especially for texture-sensitive tasks._
>
> __Scaling Synthetic Data for Image Classification:__ For classification, we took the 20K real data baseline and augmented it with increasing fractions of synthetic data : 5K, 10K, and 20K (reported). To observe the effect of generative quality, we compared a low-fidelity model (SD V1-4) with a high-fidelity model (Sana). Results: Overall, we observe that the collective average AUC increases with increasing number of synthetic samples (5K -> 10K -> 20K). This phenomenon is consistent for both generative models (Sana and SD V1-4). However, the increase in classification performance is more significant in the case of Sana (0.705 -> 0.725 -> 0.76). In the case of SD V1-4 (lower fidelity generative model), the increase in classification performance is less substantial (0.739 -> 0.751 -> 0.76).
>
> |Model  |Real + Syn.|Atelec.|C.Megaly|Consol.|Edema|En. C.|Frac.|LL  |LO  |NF  |PE  |PO  |PN  |PT  |SD  |Average|
> |-------|-----------|-------|--------|-------|-----|------|-----|----|----|----|----|----|----|----|----|-------|
> |Sana   |20K + 5K   |0.67   |0.69    |0.7    |0.79 |0.62  |0.59 |0.61|0.68|0.83|0.81|0.71|0.61|0.73|0.83|0.705  |
> |Sana   |20K + 10K  |0.7    |0.72    |0.72   |0.82 |0.63  |0.62 |0.63|0.7 |0.83|0.83|0.73|0.64|0.74|0.84|0.725  |
> |Sana   |20K + 20K  |0.76   |0.78    |0.74   |0.87 |0.66  |0.63 |0.7 |0.73|0.85|0.86|0.77|0.69|0.78|0.86|0.76   |
> |       |           |       |        |       |     |      |     |    |    |    |    |    |    |    |    |       |
> |SD V1-4|20K + 5K   |0.74   |0.75    |0.73   |0.84 |0.64  |0.59 |0.73|0.72|0.83|0.84|0.72|0.68|0.74|0.8 |0.739  |
> |SD V1-4|20K + 10K  |0.74   |0.76    |0.75   |0.86 |0.65  |0.59 |0.73|0.73|0.84|0.84|0.72|0.7 |0.76|0.85|0.751  |
> |SD V1-4|20K + 20K  |0.76   |0.78    |0.74   |0.87 |0.65  |0.6  |0.73|0.73|0.85|0.85|0.74|0.7 |0.77|0.87|0.76   |
>
> __Scaling Synthetic Data for Image Segmentation:__ For segmentation, which is highly sensitive to microscopic texture, we expanded our experiment to include all evaluated models. We compared our original augmented setting (3K Real + 3K Synthetic) against a new scaled-up setting (3K Real + 7K Synthetic). Results: For low-fidelity models (which suffer from "waxy" artefacts and poor microscopic texture), adding more synthetic data actively harms the downstream model. For example, scaling SD V1-4 from 3K to 7K synthetic samples drops the DICE score from 0.593 down to 0.575. Conversely, for high-fidelity models (Sana, Pixart Sigma, RadEdit), adding more synthetic data improves the performance, safely exceeding the real-data baseline (0.669).
>
> |Model   |3K Real, 3K Syn.|3K Real, 7K Syn.|
> |--------|----------------|----------------|
> |SD V1-4 |0.593           |0.575           |
> |SD V1-5 |0.586           |0.566           |
> |SD V2   |0.537           |0.512           |
> |SD V2-1 |0.541           |0.537           |
> |RadEdit |0.667           |0.671           |
> |Pixart Σ|0.671           |0.679           |
> |Sana    |0.667           |0.679           |
> |SD V3-5 |0.612           |0.601           |
> |Lumina 2|0.581           |0.556           |
> |Flux.1  |0.562           |0.531           |
> |LLM-CXR |0.642           |0.633           |
>
> __Action for Revision:__ We will include these new experiments, tables, and discussions in the revised manuscript. Specifically, we will add a separate section on scaling the synthetic data and its effect on downstream performance in the Appendix.

---

> > ### Author Response · Authors · 2026-04-14
> > **Response to Reviewer eKwP (3/3)**
> >
> > 5. __When Does Re-Identification become Problematic?:__ We appreciate the opportunity to clarify this crucial point. Our benchmarking on MIMIC-CXR is designed to act as a __proxy for real-world clinical scenarios__ where the data is entirely private, such as within hospitals and other clinical institutions. Furthermore, utilizing MIMIC-CXR to evaluate memorization and privacy risks aligns with established protocols in several recent privacy-focused studies (Dutt et al.,2025a; Fernandez et al., 2023; Akbar et al., 2025).
> > Our goal with CheXGenBench is to evaluate the inherent memorization capacity and re-identification risks of these generative architectures in a standardized environment. We use public data (MIMIC-CXR) to ensure reproducibility, but the re-identification risks we highlight become highly problematic when these models are deployed in real-world clinical settings. The real-world threat model is as follows: If a hospital trains a generative model on its private patient data and publicly shares the model or its synthetic outputs, a memorization-prone model allows attackers to extract exact replicas of real patient radiographs. This constitutes a severe, actionable privacy breach.
> >
> >     __Action for Revision:__ To ensure this critical context is clear to all readers, we will add a dedicated paragraph to the introduction of Section 2.3 to explicitly outline the real-world threat model. We will clarify that MIMIC-CXR serves purely as a standardized proxy for this private data to evaluate the inherent memorization risks of the architectures themselves.

---

> > > ### Comment · Reviewer_eKwP · 2026-04-30
> > >
> > > I would like to thank the authors for a detailed response. In my view, the submission is suitable for TMLR, so I recommend acceptance.

---

### Review · Reviewer_hTQJ · 2026-04-02

**Summary Of Contributions:**

This paper introduces CheXGenBench, a unified evaluation framework for chest radiograph generation problems. Building on the MIMIC-CXR dataset, CheXGenBench evaluated top-performing T2I models using 20+ commonly used metrics across Fidelity, Privacy, and Utility. Through a comprehensive evaluation, this paper also reveals critical issues, such as the fact that existing T2I models may generate re-identifiable radiographs.

**Audience:**

Yes

**Audience Explanation:**

1. This is timely research, as data generation has been widely recognized as data augmentation and has the potential to solve privacy issues.
2. As mentioned in the paper, the chest radiograph generation community still lacks a shared evaluation framework. CheXGenBench fills this gap.

**Claims And Evidence:**

Yes

**Claims Explanation:**

1. The paper discusses the three key aspects of chest radiograph generation problems and carefully picks appropriate evaluation metrics for each aspect.
2. The experiment is comprehensive, covering 11 popular T2I models using the designed benchmark.
3. The detailed analysis also identifies new potential research directions that could be further explored. (i.e., design models that generate high-fidelity images while protecting privacy).
4. The paper is well written and easy to follow.

**Requested Changes:**

1. None of the evaluated T2I models seems to be foundation models and further finetuned on the MIMIC-CXR dataset. As CheXGenBench focuses on, I would suggest providing additional evaluation of modified models tailored to radiograph generation. A comparison with evaluation methods with the original evaluation could further enhance the importance of the proposed benchmark.
2. As mentioned in the paper, existing generative models suffer from long-tail problems. Does the generation distribution follow the original distribution in terms of the category of radiograph? To the best of my knowledge, generative model tent to favor generation in the most dense region. Additional discussion or metrics on this topic could further enhance the paper's claim of the long-tail evaluation.
3. Minor: I would suggest changing the (1) (2) (3) in the last paragraph of the introduction to normal bolded text for better readability.

---

> ### Author Response · Authors · 2026-04-14
> **Response to Reviewer hTQJ**
>
> We sincerely thank Reviewer hTQJ for their valuable comments and for identifying our work as __"timely research"__ that __"fills a gap in the chest radiograph generation community"__. We address the concerns and requested changes below:
>
> 1. We respectfully want to clarify our experimental setup, as we actually did exactly as suggested. Our evaluation framework comprises both foundation models fine-tuned on MIMIC-CXR and existing domain-specific models. Specifically, we took several major foundation models from the natural imaging domain (including Stable Diffusion V1-V3, Flux, Lumina, PixArt Sigma, and Sana) and fine-tuned them entirely or via LoRA on the MIMIC-CXR dataset, as detailed in Section 2.1 and Appendix A.5. Furthermore, to provide the exact comparison mentioned in the review, we also evaluated models already tailored for radiograph generation (RadEdit and LLM-CXR) out-of-the-box as baselines.
> __Action for Revision:__ We will revise the introduction and the beginning of Section 3 to state this setup explicitly to enhance the understanding of future readers.
>
> 2. We agree with the reviewer's assessment that generative models may exhibit bias toward the most dense regions of the data distribution, and we have mathematically demonstrated this exact phenomenon in our paper.
>     * __Metrics for Mode Coverage:__ To explicitly capture how well models capture the original distribution rather than just the dense regions, we moved beyond standard FID metric and incorporated Precision, Recall, Density, and Coverage (PRDC) metrics. As discussed in Section 2.2, Recall measures how well the real distribution is captured, and Coverage determines the proportion of true data modes successfully generated. These metrics directly penalize models that only generate majority classes. Furthermore, we have explicitly stated the lack of mode coverage metrics in existing studies as a major limitation that has motivated this benchmark.
>     * __Distribution Analysis:__ We conducted a deep dive into how generation quality aligns with the original training distribution in Appendix A.3. In Table 9, we ranked each pathology by its occurrence frequency in the MIMIC-CXR training dataset and compared it to the generative fidelity (FID) for that specific pathology and found a remarkably strong positive correlation coefficient of 0.947 between the prevalence of a disease in the training data and the model's ability to generate it.
>     * That said, please recall that the models we study are prompt-conditional, so distribution in terms of simple "category of radiograph" is user-controllable, and thus straightforward to align with the training distribution. This is why we focus on the more nuanced analysis above.
>
>     __Action for Revision:__ The reviewer’s insight perfectly aligns with our findings. To further enhance the visibility of, we add a summarized version of the correlation analysis from Appendix A.3 in the main text of Section 3.1.
>
> 3. Thank you for the suggestion. We will update the formatting in the introduction to use standard bolded text to improve readability. Additionally, we will also remove similar formatting from the abstract of the manuscript.

---

### Decision · Action_Editor_G7Um · 2026-05-12

**Recommendation:** Accept with minor revision

**Additional Comments:**

The authors are suggested to complete a few minor revisions. A substantive Broader Impact Statement should be written in full, covering the risks of isolated fidelity optimization and regulatory considerations for synthetic medical data sharing. The discussion on potential pre-training data contamination should also be added. The methodology section should include a brief discussion on the framework's robustness for smaller datasets, and the Discussion or Conclusion should be expanded to propose more complex multimodal diagnostic tasks as explicit future directions.

**Audience:**

Yes

**Audience Explanation:**

Researchers working on generative models, medical AI, and privacy will find the findings directly relevant.

**Claims And Evidence:**

Yes

**Claims Explanation:**

The claims are well-supported by comprehensive experiments across 11 models and 20+ metrics. Key findings, such as the correlation between disease prevalence and generative fidelity, and the re-identification risk matching real data, are backed by quantitative evidence. Reviewer rebuttals further strengthened the evidence with additional scaling experiments and privacy analysis.

---

> ### Author Response · Authors · 2026-05-23
> **Response to Action Editor Decision and Summarizing Changes for the Revised Version**
>
> Dear Action Editor,
>
> Thank you for your time, your guidance throughout the review process, and the positive decision regarding our manuscript. We also extend our sincere gratitude to the reviewers for their constructive feedback, which has been invaluable in strengthening the rigour and clarity of our work.
>
> We have carefully addressed all remaining points raised in your decision message and the final reviewer comments. To facilitate a straightforward review of the revised manuscript, __all modifications and additions have been highlighted in red text__.
>
> Below is a summary of the key revisions incorporated into this revised version:
>
> 1. __Broader Impact Statement:__ We included a dedicated Broader Impact section that explicitly warns against the diagnostic hazards of isolated fidelity optimization, particularly the risk of reinforcing biases toward common conditions, and discusses the pressing need for regulatory frameworks to protect vulnerable patient populations.
>
> 2. __Expanded Future Directions:__ We added a dedicated paragraph in the Conclusion explicitly proposing the integration of more complex, multimodal diagnostic tasks (e.g., precise pathology localization via bounding boxes, medical VQA, and open-ended report generation) as critical future directions for the benchmark.
>
> 3. __Data Contamination & Limitations:__ We included a new dedicated discussion on limitations, explicitly addressing the nuances of foundation model pre-training, the protection afforded by MIMIC-CXR's gated PhysioNet access, and the empirical lack of zero-shot generalization as evidence against direct contamination.
>
> 4. __Methodology Robustness & Generalizability:__ We added a brief discussion in the Methodology section clarifying our dataset scale. Specifically, we noted that utilizing robust, domain-specific feature extractors (e.g., RadDino, BioViL-T) ensures the reliability, stability, and generalizability of our evaluation framework, even when operating on smaller samples of data.
>
> 5. __Expanded Dataset & Experimental Details:__ We added explicit details in Section 3 regarding the MIMIC-CXR dataset, the use of LLaVA-Rad enhanced annotations, and our specific data release plans. We also clarified the held-out test set size (5,034 samples) and justified our 20-epoch training schedule based on validation saturation in Section 2.4.
>
> 6. __Deepened Privacy Risk Analysis:__ We expanded Section 3.2 to introduce a real-data baseline Re-ID score (0.59) for physical context, demonstrating that the average synthetic image carries a similar privacy risk as authentic clinical scans. We also integrated an analysis showing that re-identification risks are notably higher for pathological scans compared to healthy ones.
>
> 7. __Texture-Topology Gap & Correlation:__ We expanded Section 4 to explicitly frame the Texture-Topology Gap as an open algorithmic challenge for the field. Additionally, we integrated a summary of our correlation analysis into Section 3.1, demonstrating the strong relationship (r=0.947) between pathology prevalence in the training data and generative fidelity.
>
> 8. __Scaling Synthetic Data Experiments:__ We added a new section to the Appendix detailing our ablation experiments, which compare how downstream utility for classification and segmentation scales with varying volumes of synthetic data across both high-fidelity (Sana) and low-fidelity (SD V1-4) models.